# Source Sector and Region Contributions to Black Carbon and PM$_{2.5}$ in the Arctic

Negin Sobhani[1,2], Sarika Kulkarni[2,3], and Gregory R. Carmichael[2]

[1] National Center for Atmospheric Research, Boulder (NCAR), Colorado, USA

[2] Center for Global and Regional Environmental Research (CGRER), University of Iowa, Iowa City, Iowa, USA

[3] California Air Resources Board (CARB), Sacramento, California, USA

*Correspondence to*: Negin Sobhani (negins@ucar.edu)

## Abstract

The impacts of BC and PM$_{2.5}$ emissions from different source sectors (e.g. transportation, power, industry, residential, and biomass burning) and geographic source regions (e.g. Europe, North America, China, Russia, Central Asia, South Asia, and the Middle East) to Arctic BC and PM$_{2.5}$ concentrations are investigated through a series of annual sensitivity simulations using the WRF-STEM modeling framework. The simulations are validated using observations at two Arctic sites (Alert and Barrow), IMPROVE surface sites over the US, and aircraft observations over the Arctic during spring and summer 2008. Emissions from power, industrial, and biomass burning sectors are found to be the main contributors to the Arctic PM$_{2.5}$ with contributions of ~30%, ~25%, and ~20% respectively. In contrast, the residential and transportation sectors are identified as the major contributors to Arctic BC with contributions of ~38% and ~30%. Anthropogenic emissions are the most dominant contributors (~88%) to the BC surface concentration over the Arctic annually; however, the contribution from biomass burning is significant over the summer (up to ~50%). Among all geographical regions, Europe and China have the highest contributions to the BC surface concentrations with contributions of ~46% and ~25% respectively. Industrial and power emissions had the highest contributions to the Arctic sulfate (SO$_4$) surface concentration with annual contributions of ~43% and ~41% respectively. Further sensitivity runs show that among various economic sectors of all geographic regions, European and Chinese residential sectors contribute to ~25% and ~14% of the Arctic

average surface BC concentration. Emissions from Chinese industry sector and European power sector contribute ~12% and ~18% of the Arctic surface sulfate concentration. For Arctic $PM_{2.5}$, the anthropogenic emissions contribute > ~75% at the surface annually, with contributions of ~25% from Europe and ~20% from China; however, the contributions of biomass burning emissions are significant in particular during spring and summer.

The contributions of each geographical region to the Arctic $PM_{2.5}$ and BC vary significantly with altitude. The simulations show that the BC from China is transported to the Arctic in the mid-troposphere, while, BC from European emission sources are transported near the surface under 5km, especially during winter.

## 1. Introduction

Arctic temperature has increased more than the mean global surface air temperature over the past century

due to various positive feedbacks and amplification mechanisms such as albedo feedback caused by black carbon (BC) deposition (AMAP, 2011a, 2011b, 2015; Cohen et al., 2012; IPCC, 2013; Screen and Simmonds, 2010). Long-range transport of atmospheric particulate matter (PM) from mid-latitudes to the Arctic is the main contributor to the Arctic aerosol load (AMAP, 2011b; Law and Stohl, 2007; Quinn et al., 2007). Several studies, as early as the 1980s, reported a distinctive seasonal cycle in the Arctic aerosol and BC concentration and visibility

(Barrie, 1986; Quinn et al., 2007; Rosen et al., 1981; Schnell, 1984; Wang et al., 2011). The so-called Arctic Haze phenomenon in the winter-spring period has been attributed to increased levels of transported PM from anthropogenic emission sources at lower latitudes and slower wet deposition removal processes (Barrie et al., 1981; Law and Stohl, 2007; Quinn et al., 2002, 2007).

BC is a critical component of the Arctic haze, and influences global climate and water cycles in various

ways (AMAP, 2011b; Bond et al., 2013; Shindell et al., 2008). BC particles in the atmosphere absorb solar radiation and warm the surrounding air. When deposited on snow and ice, BC reduces the surface albedo and absorbs more solar radiation; hence, increases the temperature of snow and accelerates the snow melting process (Clarke and Noone, 1985; Flanner et al., 2007; Hansen and Nazarenko, 2004; Koch et al., 2007; Wiscombe and Warren, 1980). Although BC is a minor contributor to aerosol loading (~10%), it has been identified as the second largest

contributor to global warming after carbon dioxide ($CO_2$) (Bond et al., 2013; Ramanathan and Carmichael, 2008). Studies suggest that BC has caused ~25% of the 20[th] century warming over the Arctic (Bond and Sun, 2005; Koch and Hansen, 2005; Ramanathan and Carmichael, 2008). Although BC plays a significant role in global climate, there is high uncertainty in assessing the magnitude of BC effects on radiative forcing and climate (Bond et al., 2013; Flanner et al., 2007). Considering the short atmospheric lifetime of BC and its significant impact on the

Arctic climate, mitigating BC emissions provides us with an opportunity to decrease BC concentration in the atmosphere immediately to reduce near-term climate impacts (Bond and Sun, 2005; Hansen and Sato, 2001; Jacobson, 2002; Ramanathan and Carmichael, 2008). To devise effective global BC emission abatement policies, it is necessary to quantify the contribution of each geographical source region and source sector, and to identify the major transport pathways to the Arctic (AMAP, 2011b).

BC in the Arctic has both natural (e.g. biomass burning) and anthropogenic sources (e.g. smelter emissions from Norilsk or the Kola peninsula) (Schmale et al., 2011), but there are very few emission sources locally within the Arctic region (AMAP, 2011b; Law and Stohl, 2007). Hence, the main contributor to BC in the Arctic atmosphere is the long-range transport of particles from mid and high-latitude regions (AMAP, 2011b; Bond et al., 2013; Law et al., 2014; Law and Stohl, 2007). Several studies have shown that transport of aerosols from mid-latitudes is the most significant transport mechanism to the Arctic pollution (AMAP, 2011b; Law and Stohl, 2007). Previous studies in literature have identified Europe as the major source region contributor to the Arctic BC concentrations (Barrie, 1986; Quinn et al., 2007, 2008; Raatz and Shaw, 1984; Shaw, 1995). However, during the past two decades emissions from East Asia have increased rapidly due to the vast economic growth, while the emissions from Europe have declined during the same time period (Streets et al., 2009). Recent studies have shown the significant contribution of Asian emissions to the Arctic, especially during winter-spring (Breider et al., 2014; Fisher et al., 2010; Ikeda et al., 2017; Koch and Hansen, 2005; Sharma et al., 2013a; Shindell et al., 2008; Stohl, 2006; Wang et al., 2014, 2011). However, there are significant uncertainties associated with these estimates (Fisher et al., 2010; Koch and Hansen, 2005; Sharma et al., 2013b; Stohl et al., 2006; Wang et al., 2011), due to uncertainties in emissions, and the complicated transport pathways from mid-latitudes to the Arctic (Bian et al., 2013; Fuelberg et al., 2010).

Modeling BC concentrations over the Arctic is considered a challenging task for chemical transport models (Eckhardt et al., 2015; Koch et al., 2009; Shindell et al., 2008; Wang et al., 2011). Previous model inter-comparison studies have shown an order of magnitude differences between observation and model (Bond and Sun, 2005; Wang et al., 2011). The studies by Shindell et al., 2008 and Koch et al., 2009 have shown negative bias between model and observation. However, Shwartz et al. 2010 shows positive bias comparing global models with observation (Shwartz et al. 2010, Sharma et al., 2013). These differences between model performances are primarily due to the high uncertainties in emissions, Arctic meteorology, and scavenging efficiency for calculating wet deposition (Bourgeois and Bey, 2011; Browse et al., 2012; Eckhardt et al., 2015; Garrett et al., 2010, 2011, Liu et al., 2015, 2011; Marelle et al., 2017; Qi et al., 2017a). The modeling inter-comparison study by Eckhardt et al., 2015 showed

that current models (including both atmospheric chemistry transport and climate models) were unable to reproduce the observed BC seasonality at the surface. There are also high discrepancies among different models in capturing BC concentrations over the Arctic (Eckhardt et al., 2015; Shindell et al., 2008), which is caused by various factors including emissions, meteorology, and transport patterns.  The uncertainties associated with emissions is a key component of this inter-model variability and differences between simulations and observations. According to Ramanathan and Carmichael, 2008, regional emissions can have a factor of 2 to 5 uncertainty. For example, while previous studies estimated that oil and natural gas flaring is an important sector contributor to the Arctic (AMAP, 2015; Eckhardt et al., 2015; Huang et al., 2014, 2015; Sand et al., 2015; Stohl et al., 2013; Xu et al., 2017), a recent paper (Winiger et al., 2017)  showed that emissions from oil and gas flaring contribute to only ~6% of  Arctic BC concentration indicating a 6.25× overestimation of flaring emissions in the previous studies.

Other factors that also contribute to the model-observation offset in the Arctic region are the uncertainties and errors in meteorology and transport mechanism (Jiao et al., 2014). Finally, the representation of the particle processes in the atmosphere is another major source of uncertainty in the inter-model variability. Errors and uncertainties in dry and wet removal processes (including in-cloud and below-cloud mechanisms) at high altitudes is a major source of uncertainty. Mahmood et al., 2016 study indicates that scavenging of BC in convective clouds outside the Arctic, substantially influences BC vertical distributions and overall wet deposition efficiency within the Arctic; hence, is one of the major cause of discrepancies in Arctic BC burdens among different models used in Eckhardt et al., 2015. Marelle et al., 2017 indicates that both surface and tropospheric BC in the Arctic are highly sensitive to the representation of cumulus cloud processes impacting aerosols.

It is crucial to identify the sources of Arctic pollution in order to devise effective control strategies for mitigating the Arctic air quality, climate change, and radiation imbalances. The primary goal of this study is to quantify the relative contributions of different source sectors and source regions to the arctic aerosol concentration (surface and column abundances) and its impact on Arctic air quality through a series of model sensitivity simulations. Although the aerosol vertical profiles and column abundances are discussed, addressing the aerosol radiative and climate impacts is beyond the scope of this work. In this study, we designed a modeling framework (WRF-STEM) for analyzing BC, organic carbon (OC), sulfate ($SO_4$), $PM_{2.5}$, and $PM_{10}$ concentrations over the Arctic from April 2008 to July 2009. We utilize this system to study the seasonal variations in the contributions of emissions from different source sectors (e.g. transportation, power, industry, residential, and biomass burning) and geographic source regions (e.g. Europe, North America, China, Russia, Central Asia, South Asia, and the Middle

East) on Arctic PM mass concentration (Figure SM1). Section 2 describes the data sources and modeling framework utilized in this study, while the findings are discussed in section 3 followed by conclusions in section 4.

## 2. Method and Data

### 2.1. Modeling System

#### 2.1.1. Meteorological Model

The Weather Research and Forecasting model (WRF) (Skamarock et al., 2008) version 3.4 model was used for producing necessary meteorological inputs for the STEM model. The ice sheet coverage, initial and boundary conditions for the model were provided by the National Center for Environmental Prediction (NCEP) Final Analysis (FNL, http://rda.ucar.edu/datasets/ds083.2/). The meteorological factors affecting chemical distribution and concentration were imported into the STEM model every 6 hours as described in Kulkarni et al. 2015 and Sobhani, 2017.

#### 2.1.2. Emissions

The base emission setup used for this modeling study is similar to Kulkarni et al. 2015, except that anthropogenic emissions were updated to the Hemispheric Transport of Air Pollution phase 2 emissions inventory (HTAP_v2.2) for the year 2008 (Janssens-Maenhout et al., 2015). HTAP_v2.2 emission inventory contains comprehensive harmonized sector-specific $0.1° \times 0.1°$ longitude-latitude emission grid maps for $SO_2$, $NO_x$, NMVOC, $NH_3$, $PM_{10}$ , $PM_{2.5}$, BC, and OC with monthly and yearly temporal resolution for the years 2008 and 2010 (Janssens-Maenhout et al., 2015, data available at http://edgar.jrc.ec.europa.eu/htap_v2/index.php?SECURE=123). For this study, we utilized the monthly-varying emissions available for 2008 from HTAP v2.2 emission inventory. HTAP_v2.2  emission is based on a collection of different regional gridded emission inventories per sector and per region including that of the European Monitoring and Evaluation Programme (EMEP) and Netherlands Organisation for Applied Scientific Research (TNO) for Europe,  the Environmental Protection Agency (EPA) for USA, the EPA and Environment Canada (for Canada), and the Model Inter-comparison Study for Asia (MICS-Asia III) for China, India and other Asian countries (Janssens-Maenhout et al., 2015; Li et al., 2017; Lu et al., 2011). For the rest of the world (South America, Africa, Russia, and Oceania) the emission grid maps of the Emissions Database for Global Atmospheric Research

(EDGARv4.3) bottom-up inventory was used in HTAP_v2.2 (Janssens-Maenhout et al., 2015). The sectors in HTAP_v2.2 emission dataset are based on IPCC 1996 categories definitions. In this study, Energy (alternatively named as Power) sector is defined as total emissions from stationary and mobile energy activities for electricity generation, which includes fuel combustion as well as fugitive fuel emissions. The industrial sector includes emissions from industrial large-scale combustion emissions other than electricity productions (power sector) and emissions from industrial processes and solvent productions and applications. Emissions from the residential sector are from small-scale combustion including heating, cooling, illuminations, cooking, and other auxiliary engines (such as lifting systems) to equip residential buildings, commercial buildings, agricultural facilities (including fisheries), waste-water treatment, and solid waste disposal and incineration.

Emissions from the residential sector have strong seasonal (monthly) variations, which is negatively correlated to the temperature in most of the regions due to the use of heating systems (Janssens-Maenhout et al., 2015). In some developed countries, the residential sector emissions have a positive correlation with the temperature during the summer due to the increase in emissions from air conditioning devices (Janssens-Maenhout et al., 2015). Emissions from transport, industry, and energy sectors show modest seasonality in all regions (Janssens-Maenhout et al., 2015). There are high uncertainties in HTAP v2.2 $PM_{2.5}$ and BC emissions emerging from different sources (especially transport and residential sectors). These uncertainties originate the uncertainties in officially announced annual inventories provided by countries, uncertainties due to process representation (the quality and representativeness of the controlled emission factors), and uncertainties due to aggregations (grid maps used for allocating national totals for a source category will be different from the maps used at national levels). It is important to note that $PM_{2.5}$, BC, and OC emissions from residential and transport sectors are qualitatively classified as highly uncertain in HTAP v2.2.

A new source category of emissions from open waste burning from Wiedinmyer et al., 2014 were also utilized in this study. For carbonaceous aerosols and PM emissions from biomass burning sector, the Fire Inventory from NCAR (FINN v1) emissions from Wiedinmyer et al., 2011 was used. FINN provide daily global emissions based on satellite (e.g., MODIS) observation for detecting active fires as thermal anomalies and land cover change (Wiedinmyer et al., 2011). Dust emissions were estimated using Uno et al., 2004 method for grids with snow cover <1%. Further details of the biomass burning and dust emissions described in Kulkarni et al. 2015 and Sobhani, 2017.

Figure 1 shows the regional distribution of anthropogenic and wildfire BC emissions for the modeling domain. The major anthropogenic BC emission hotspots are over China and India along with significant emissions from Eastern Contiguous United States (CONUS), Europe, and Northern Middle East regions. The major hotspots of wildfire BC emissions are over South East Asia, Siberia, and Europe. There are also wildfire emission sources from Southeastern and Western CONUS (Figure 1).

Figure 2 shows the anthropogenic BC emissions from different economic sectors. The residential source sector is the primary source of BC emissions over Asia (including China, India, and Southeastern Asia) with values ranging from ~45% to ~95% of total anthropogenic BC emissions. The transportation sector is the dominant emission sector over North America and Central Asia with values ranging from ~35% to ~90%. The industry sector contributes between ~35% to ~50% of total BC emissions over Central Asia and Siberia.

### 2.1.3. Chemical Transport Model

The WRF-STEM modeling framework is similar to that used by Kulkarni et al. 2015 except for updated anthropogenic emissions (described above). The STEM model is a regional scale Chemical Transport Model (CTM) developed at the University of Iowa in the 1980's (Carmichael and Peters, 1984, 1986) and has been continuously developed since then. The STEM model includes emission, transport (convective and diffusive), and deposition of particles and chemicals based on an Eulerian approach. This model is investigating the convective-diffusion equation below with Eulerian approach to calculate the concentration of chemical species i ($c_i$).

$$\frac{\partial c_i}{\partial t} + \nabla(v c_i) = \nabla . K . \nabla c_i + R_i + S_i + G_i$$

In the above equation, $c_i$ is the gas phase concentration of compound i, $v$ is the wind velocity vector, K is for eddy diffusity tensor, $R_i$ is the total reactions of species i , $S_i$ denotes the sources for species i and $G_i$ is the mass transfer between gas and liquid (Kulkarni, 2009). The dry deposition of particles was calculated based on the resistance in series parameterization developed by Wesely and Hicks, 2000 and the values vary with meteorological conditions and land cover (Adhikary et al., 2007; Kulkarni et al., 2015; Sobhani, 2017; Uno et al., 2004). Wet deposition was modeled as a function of loss rate based on the meteorological fields (precipitation rates) from the WRF model as described in Uno et al., 2003 and Adhikary et al., 2007. Aging has been considered for both BC and OC particles using 7.1e-6 s$^{-1}$ as the aging rate (Adhikary et al., 2007; Cooke and Wilson, 1996). In this study, we used STEM model for simulating BC, OC, sulfate ($SO_4$), $SO_2$, $PM_{2.5}$, $PM_{10}$ , and other primary emitted $PM_{2.5}$ and $PM_{10}$ .

The modeling domain for both WRF and STEM models covers most of the Northern hemisphere including the significant emission sources such as Asia, Russia, Europe, and North America. Also, the model extends over the Northern Africa, Middle East, and South Asia to include the dust emissions from the arid regions and anthropogenic emissions from the population-dense regions. The model used a polar stereographic map projection with 60 km horizontal resolution (249×249 grid cells). Regional models such as STEM require initial and boundary conditions from a larger-scale model to achieve reasonable predictions (Abdi-Oskouei et al., 2018; Tang et al., 2007). The STEM model used fixed boundary conditions for these annual simulations. The boundary conditions varied spatially and vertically based on observations from previous aircraft field experiments and discussed in detail in Tang et al., 2004. Further details describing this modeling system can be found in Kulkarni et al., 2015 and D'Allura et al., 2011.

### 2.1.4. Sensitivity Simulations

For making effective emission mitigation policies, it is essential to assess the impacts of source sectors and source regions on the Arctic pollution. The base simulations and sensitivity analysis with perturbed emissions were performed to quantify the impacts of various emission sectors and regions on the concentrations of PM and its components in the Arctic. The sector contributions were calculated using a series of model runs by eliminating the emissions of a particular sector each time. The base simulation included emissions from all sectors and used meteorology from the WRF model for the study period. The contributions of each sector to the PM concentrations were calculated as the difference between the base case and a simulation including all emissions but zeroing out the specific sector. Additional simulations were performed to calculate the source contribution from specific regions to PM concentrations over the Arctic. Using a similar method, sensitivity simulations were also performed to estimate the contributions of economic sectors from each of the geographic source regions to the Arctic surface and column concentrations. These large emissions changes can lead to errors in secondary pollutants if the chemistry is non-linear. As BC, dust, and primary PM are primary species the results are not sensitive to non-linear effects. Sulfate is a secondary pollutant but its chemistry (in cloud and gas phase) is treated as a linear process in this model experiment.

### 2.2. Observations

The modeling system performance was evaluated by comparing simulated values with aircraft observations from the National Aeronautics and Space Administration (NASA) Arctic Research of the Composition of the Troposphere from Aircraft and Satellites (ARCTAS) field campaign (Jacob et al., 2010). The ARCTAS field

campaign measurements included observations from DC-8, P-3, and B-2000 research aircraft and data analysis and forecasts by different global and regional modeling teams. The ARCTAS field campaign took place as a part of the international POLARCAT framework (POLar study using Aircraft, Remote sensing, surface measurements and models, of Climate, chemistry, Aerosols, and Transport; see Law et al., 2014 and www.polarcat.no ) during the 2007-2008 international polar year, with the goal to better understand the factors causing changes in the Arctic atmospheric composition and radiative forcing (Jacob et al., 2010; Law et al., 2014). The spring phase (ARCTAS-A) which happened during April 2008, was concurrent with an unusually higher number of Siberian fires, which subsequently caused higher concentrations of carbonaceous aerosols (Fuelberg et al., 2010; Jacob et al., 2010; Kondo et al., 2011; Liu et al., 2015; Matsui et al., 2011; McNaughton et al., 2011; Spackman et al., 2010; Wang et al., 2011; Warneke et al., 2009). Figure SM2 panels show the flight pathways of all ARCTAS flights during spring (ARCTAS-A) and summer 2008 (ARCTAS-B) respectively.

The model performance was evaluated during different seasons by comparing simulated concentrations with the surface observations at two sites located in the Arctic: Barrow, Alaska (156.6° W, 71.3° N, 11 m a.s.l.) and Alert (Nunavut), Canada (62.3° W 82.5° N, 210 m a.s.l.) depicted in Figure SM2. The Barrow site is located northeast of the Barrow town at the northern edge of Alaska. Observations at Barrow are retrieved from National Oceanic and Atmospheric Administration (NOAA) Global Monitoring Division (GMD), where a particle soot absorption photometer (PSAP) is used for measuring BC light absorbing coefficient at three wavelengths (476, 530, and 660 nm) (Bodhaine, 1989; Bond et al., 1999; Delene and Ogren, 2002; Data is available at https://esrl.noaa.gov/gmd/aero/net/). The Alert station, located in the northernmost Qikiqtaaluk region of Canada, is mostly isolated from both local and continental source regions. The Alert observatory is the most northerly site of World Meteorological Organization (WMO) Global Atmosphere Watch (GAW) network. Alert BC concentrations are calculated using light absorption coefficient data measured by Environment and Climate Change Canada using a PSAP (Radiance Research, Inc.) at three wavelengths (476, 530, and 660 nm) (Sharma et al., 2004, 2013a, 2017; Data is available at http://ebas.nilu.no/). The light absorption coefficients are converted to Equivalent Black Carbon (EBC) using mass absorption cross-section (MAC). In this study for calculating EBC concentration, light absorption coefficient at 530 nm was used with a MAC value of 9.5 $m^2$/g as recommended by McNaughton et al., 2011; Qi et al., 2017a; Sharma et al., 2013b; Stohl et al., 2013; Wang et al., 2011.

Sulfate measurements at Barrow and Alert are taken using ion chromatographic analysis (Hirdman et al., 2010a; Quinn et al., 1998; Sirois and Barrie, 1999). A basic high volatile sampler from Sierra Instruments is used for collecting aerosol samples at both the monitoring sites. The measured sulfate concentrations at both Alert and Barrow sites were corrected by subtracting sea salt component using aerosol sodium (Na+) and chlorine (Cl-), which are mostly from marine sources (AMAP 2015; Barrie and Hoff, 1985; Hirdman et al., 2010; Quinn et al., 1998, 2000). Therefore, the reported non-sea salt (nss) sulfate can be directly compared with the modeled values. It should also be noted that the sample durations for Alert and Barrow sites varied 1-5 days for Barrow and 3-9 days for Alert (Hirdman et al., 2010a; Quinn et al., 1998; Sirois and Barrie, 1999). For further validating the model's performance outside the Arctic's circle, BC surface concentration data was evaluated using annual average data from 168 Interagency Monitoring of Protected Visual Environment (IMPROVE) sites over North America and is described in section 3.1.2.

## 3. Results and Discussions

### 3.1. Model Evaluation

#### 3.1.1. Meteorological Model Evaluation

Since meteorology drives the underlying transport patterns in air quality simulations, WRF model performance was evaluated using observations from 2008 ARCTAS field campaign. The spring and summer 2008 ARCTAS flight tracks are illustrated in Figure SM2 a and b respectively. To evaluate performance of the model for different regions in the Arctic, the flights were categorized into the following 7 categories based on the location and time of the flights including, 1- spring Alaska local flights, 2- spring Greenland flights, 3-spring transit flights, 4- summer California flights, 5- summer Canada local flights, 6- summer Canada Greenland flights, and 7- summer transit flights. Table 1 shows the different flight categories and the date of the flights corresponding to each category. The model data were evaluated for each flight and all flight categories.

Figure SM3 boxplots compare the model with measured meteorological data for each of the flights. Each flight category is shaded with a different color and the spring and summer transition flights are not shaded. The simulated meteorological variables were extracted along the DC-8 flight pathways and compared against observational data measured on the DC-8. For each of the flights, simulation and measured data, combined at all

altitudes, were summarized into one separate box/whisker plot. Table 2 shows a statistical summary of comparisons between modeled and measured metrological parameters.

These results and further analysis by altitude (Figure SM5) show that the modeled meteorology captured the many of the observed features seen in temperature, Relative Humidity (RH), and wind speed. Temperature shows a slight positive bias for summer flights and a negative bias at higher altitudes during spring. In addition, the model underpredicts RH during the spring and summer California flights, while it overpredicted it during the summer Canada Greenland flights. The RH underprediction happens at lower latitudes for spring flights, and overprediction occurs in higher altitudes for summer flights. The model also tends to slightly overpredict wind speed by ~4% at higher altitudes during spring flights. The model underpredicted the wind speed for all summer California flights. The model captured the RH vertical distribution in the lower troposphere but displays a substantial negative bias at altitudes above ~4km. This indicates the difficulties in capturing the complex ice and cloud formation properties at high altitudes in the polar region during springtime.

Table 2 summarizes the statistical summary of the major meteorological variables for both ARCTAS observation data and model output. Based on this table and box and whisker plots analysis (Figure SM3 and SM5) the model captures vertical profiles and magnitudes of meteorological observations from ARCTAS field campaign.

### 3.1.2. Concentration Evaluation

*Concentration Evaluation along ARCTAS DC-8 flights*

The simulated air pollution concentrations were evaluated using NASA ARCTAS flight data. Figure 3 shows boxplots comparing concentrations of BC, sulfate, and $SO_2$ for model and observations for each ARCTAS flight. The flight categories are shaded similar to Figure SM3. The results show that generally, simulated BC follows the same flight-by-flight variation as observed with an overall high bias (Figure 3).The vertical distributions of aerosols play a critical role in determining the impacts of aerosols on radiative forcing. Figure 4 compares the vertical BC and sulfate concentration profiles for all flights. The vertical profiles for each flight category are shown in Figure SM4.  In the vertical profile plots, both modeled and observed values are binned by flight altitudes every 1 km. The model captured the vertical variability of BC and $SO_4$ concentration well (Figure 4). For BC, both observation and model show the highest values near the surface.  The simulated BC values are biased high above 5 km for all flight categories (Figure SM4). There is also a constant overprediction of sulfate above 5km, which may be due in part to an underprediction of RH, resulting in underestimation of wet removal and in-cloud scavenging at altitudes above 5km. Pollutant transport across the Pacific happens in discrete plumes during

Springtime (Adhikary et al., 2010). CTMs tend to disperse these plumes in vertical layers of the atmosphere too much. This spreading typically results in decreases in modeled peak values (Adhikary et al., 2009; Kulkarni, 2009). The underestimation of BC at the surface may also be attributed to an underestimation of BC emissions especially at higher latitudes e.g., gas flaring (Huang et al., 2014, 2015; Stohl et al., 2013) and shipping emissions (Marelle et al., 2016). The overprediction of BC at higher altitudes might be due in part to underestimations of BC removal by frozen clouds and precipitations (Koch et al., 2009).

### *Surface Concentration Evaluation at Barrow and Alert*

For evaluating the model performance in capturing the seasonality of BC concentration in the Arctic region, we compared the simulated BC surface concentrations with BC data available at Barrow and Alert stations for the duration of the study. When using EBC values, it is imperative to keep in mind that the MAC values used for estimating EBC has a large range (from $5 m^2/g$ to $20 m^2/g$) and EBC concentrations has at least a factor of two uncertainty (Bond and Bergstrom, 2006; Liousse et al., 1996; Sharma et al., 2002, 2017; Weingartner et al., 2003). Traditionally, a MAC value of $10 m^2/g$ was used for EBC calculations for aged BC particles and Sharma et al., 2013a used MAC values of $19 m^2/g$ for both Barrow and Alert sites. However, recent studies suggest much lower values for MAC compared to the $9.5 m^2/g$ used for this study. Sharma et al., 2017 suggest MAC values of $5 \pm 2$ $m^2/g$ for summertime and Sinha et al., 2017 suggested MAC values as low as $8.5 m^2/g$ for Barrow site. Using lower MAC values will result in higher observed EBC.

Figure 5 shows the time-series boxplots of simulated BC vs. observed EBC concentration for the duration of the study at the surface for the Alert and Barrow sites. The model was able to accurately capture the seasonality of BC in both sites. Both model and observation show higher values of BC during winter and spring, indicating the Arctic Haze. At the Alert site, the model especially captured the wintertime and springtime peak values; however, it overpredicted the summer BC concentration. Using lower MAC values as suggested by Sharma et al., 2017 for summertime results in 1.9× higher observed EBC which will be closer to the simulated values.

At Barrow site, the model consistently overestimates the BC concentrations during the year. The overestimation of BC during summer can be due to the large contributions of biomass burning from Siberia in the simulations caused by overestimations of emissions and/or too little removal during transport. Furthermore, Stohl, 2006 and Stohl et al., 2013 studies discussed that the biomass burning contributions from remote locations were unintentionally removed in the Barrow measurements data processing. By removing the data cleaning for Barrow site, the observations were increased by a factor of 2-3 during summer (Stohl et al., 2013).

Figure 6 shows monthly boxplots comparing simulated sulfate with observed values at Alert and Barrow. Both stations show strong seasonal variation with the minimum occurring during summer and early fall similar to BC. As discussed above for BC, this is due to the northward retreat of the Arctic front and efficient wet scavenging during summer. The model accurately captured the seasonality of observed sulfate at both sites. The summertime minima of sulfate reflects the less effective transport and high scavenging during summer. At Barrow, the model overpredicted the observed values throughout the year. However, during spring and winter, the simulated sulfate values are much closer to the observation. It should be noted that the observations from Barrow site has large data gaps and missing data possibly due to equipment malfunction. To avoid local contamination, the sector source controlled sampling method removes data suspected to be contaminated by the town of Barrow (Bodhaine, 1989, 1995; Fisher et al., 2011; Hirdman et al., 2010a). The significant data gaps might introduce biases in the monthly calculations. There were also no sulfate measurements for July, August, and December 2008. The model overpredicted sulfate at the Alert site, except for wintertime. During winter, the model accurately predicted the range of simulated sulfate at the Alert site. The overprediction during summer might be due to the less effective scavenging processes and higher magnitude of transport in the model. The results are similar to the Hirdman et al., 2010b study, which used nss sulfate monthly averages dung the years 2000-2006. Observations and model show that Barrow shows much higher concentrations of sulfate throughout the year when compared to the Alert site.

### *BC Concentration Evaluation for IMPROVE sites*

The simulated air pollution concentrations were further evaluated using data from 168 IMPROVE sites over the U.S. for the period of April 2008 to July 2009 (Data available from http://vista.cira.colostate.edu/improve/Data/IMPROVE/AsciiData.-asp). Figure 7 shows the annual mean surface BC concentration over the U.S. compared with observations at IMPROVE network sites. Each site is represented as a circle in the map. The average model BC over the U.S. is 0.16 $\mu g/m^3$ while average IMPROVE data is 0.19 $\mu g/m^3$. Further statistical analyses show that the root-mean-square deviation (RMSE) between model and observation is 32% and the mean bias error (MBE) is 0.03 $\mu g/m^3$.

### 3.2. Spatial Distribution of PM Species

BC and sulfate are major components of $PM_{2.5}$, and can be transported over long distances and across the continents. Both BC and sulfate have several anthropogenic and natural emission sources. Figure 8 shows the annual average concentrations of surface BC, sulfate, $PM_{2.5}$, and PM Ratio over the entire modeling domain. Figure 8-a shows that the modeled BC surface concentration is in the range of ~0.25 to ~3 $\mu g/m^3$. The major BC hotspots

are over Southeast Asia, northern India, and China with annual average concentrations of ~3 µg/m$^3$. Furthermore, the seasonal and monthly results show that BC concentration peaks during wintertime since there are higher biomass and fossil fuel burning for heating during the winter season. The annual average surface concentration over the U.S. is 0.16 µg/m$^3$ with the maximum BC over the Eastern U.S. with the average of 0.75 µg/m$^3$. The annual average BC for the Arctic region (latitudes > 60°N) is between ~0.025 µg/m$^3$–0.075µg/m$^3$ with the minimum occurring over Greenland, Alaska, and Northern Canada. The simulated annual BC average for the Arctic area (latitudes > 60°N) is on average ~ 0.065µg/m3. This value is consistent with the average of 0.06µg/m3 over the Arctic from Sharma et al. 2013-b.

Sulfate can be produced by sea spray or volcanos, but they are mostly from oxidation of $SO_2$ emitted during combustion of sulfur-containing fossil-fuels (Forster et al., 2007). Sulfate scatters solar radiation and has a negative direct radiative forcing. Figure 8-b shows that the major sulfate levels are in Asia and northern India, with less intense but significant concentrations over Europe and eastern CONUS. The concentration of sulfate particles over East Asia is approximately two times higher than sulfate concentration over the eastern CONUS and Europe. This is partly due to higher $SO_2$ emissions in the Asian region and relatively faster $SO_2$ oxidation rates (Chin et al., 2007).

Figure 8-c shows the distribution of surface PM$_{2.5}$. Major PM$_{2.5}$ hotspots are over the Persian Gulf, Central Asia, northern India, and northern Africa with annual average maxima as high as ~80 µg/m$^3$ around the Persian Gulf. The Arctic area (above 60°N) show values between 1- 5 µg/m$^3$ with maximum occurring over northern Europe and northern Russia. Greenland, Northern Canada, and Alaska show average PM$_{2.5}$ concentrations of ~2 µg/m$^3$. Figure 8-d shows the PM$_{2.5}$/PM$_{10}$ ratio, which is an indicator of relative contributions from anthropogenic and natural sources. The arid regions with high natural dust emissions such as northern African, the Persian Gulf, and Central Asia show lower PM ratios indicating the major contributions of dust to PM over these regions. Over the oceans, the PM ratio is very low (0.1-0.2) caused by higher contributions of sea salt to PM$_{10}$ and low PM$_{2.5}$ concentration (~84% contribution of coarse sea salt to PM$_{10}$ over the Atlantic Ocean and ~75% over the Pacific Ocean). Higher PM ratio values in eastern Asia and eastern CONUS indicate that the sources of PM in these regions are mostly anthropogenic.

### 3.3. Sources of Arctic PM

#### 3.3.1. Source Sectors Contributing to PM Surface Concentration

Due to the significant contribution of BC to the warming seen over the Arctic and its amplification mechanisms, it is important to understand the influence of specific source regions and source sectors on the Arctic BC concentration. Figure 9 shows the 5 major source sector contributions percentage to BC surface concentrations. Transportation is the major sector contributor over North America with contributions ranging from ~30% to ~55%. The residential sector is the major contributor to BC over China and South Asia with maximum residential contribution percentage as high as ~70 %, which is generally consistent with spatial pattern of emissions (Figure 2). However, the residential sector has a significant (~25%) contribution over Western U.S. reflecting the outflow of Asian BC over Pacific Ocean and to the West Coast. The residential, transportation and industrial sectors are the major emission sources over Europe as shown in Figure 2. Over the Arctic (60 °N and above) residential and transportation sectors show maximum contributions of ~38% and ~30%, respectively. The contribution from the biomass burning sector over the Siberian Arctic is substantial with values as high as ~40%, which can be attributed to the large number of forest fires particularly during springtime. Previously, Stohl et al., 2013 study suggested that emission from oil and natural gas flaring in Russia is an important but overlooked source of Arctic BC contributing to 66% of total anthropogenic emissions within the Arctic (latitudes > 66 °N). Similarly, Huang et al., 2015 estimated that gas flaring emissions accounts for 36.2% of total anthropogenic BC emissions from Russia. Using similar emission inventory, AMAP, 2015, Eckhardt et al., 2015, Huang et al., 2014, Huang et al., 2015, Sand et al., 2015, Stohl et al., 2013 , and Xu et al., 2017 concluded that flaring is a significant contributor to Arctic BC. However, a recent study (Winiger et al., 2017) using Bayesian approach, FLEXPART, and 2 year continuous observations identified the errors in space allocation of previous emission inventory and suggested -84% reduction of flaring emissions, which translates to (6.25x) overestimation of flaring emissions. Winiger et al., 2017 study shows that contribution of gas flaring is relatively small (~6%) compared to residential (~35%) and transport (~38%) sectors, which is similar to our results showing residential and transportation are contributing ~38% and ~30% to the Arctic BC.

Industrial and power emissions had the highest contributions to the Arctic sulfate concentration with annual contributions of ~43% and ~41% respectively, while biomass burning, power and industrial emissions have the highest contributions (~30%, ~25%, ~ 20%) to Arctic $PM_{2.5}$ (Figure SM6 and SM 7 respectively). Figure SM 6

shows the large contributions of power sector to Europe sulfate and high contributions industrial sector over North America. Based on Figure SM7, power sector is the major contributor to $PM_{2.5}$ over the Europe and eastern US, and Industrial sector is the most significant contributor to $PM_{2.5}$ over Canada, western US, Russia, and China. Biomass burning has significant contributions to $PM_{2.5}$ over southeastern Asia, Western US, and Russia. Residential sector has high impact on Eastern China and Indo-Gangetic plain $PM_{2.5}$ surface concentration based on Figure SM 7.

The seasonality in sector contributions to the Arctic pollution is shown in Figure 10. Figure 10-top panel shows the time series contribution from five emission sectors to BC surface concentration (calculated as the area average surface concentration for latitudes 60°N and above) over the Arctic. The surface concentrations range from 0.05 μg/m³ to 0.2 μg/m³ over the Arctic with the maximum values occurring during wintertime, indicating the prevalence of Arctic haze. The contribution from residential sector significantly increases during wintertime, since burning of biofuels and coal are the main heating resource at higher latitudes. Furthermore, there is a high seasonal variability in the contribution of biomass burning with maximum values occurring during the springtime due to the widespread seasonal agricultural burning over Russia and the increased occurrence of Siberian forest fires (AMAP, 2011b). During spring 2008, biomass burning was reported to be unusually high (AMAP, 2011b; Jacob et al., 2010; Liu et al., 2015; Matsui et al., 2011; Wang et al., 2011; Warneke et al., 2009). Furthermore, during the spring the Arctic front is more southerly on the Eurasian side (Bond et al., 2013; Stohl, 2006). Hence, the BC emitted from agricultural burning and boreal forests from Europe and Russia transport easily, especially at lower altitudes. These results are similar to Qi et al., 2017b , Brock et al., 2011, Warneke et al., 2010, and Bond et al., 2013 , which suggest that high-latitude agricultural and boreal forest fire is one of the main contributors to BC over the Arctic during spring 2008.

Figure SM8 shows the seasonal variation of contributions of different economic sectors to Arctic BC column concentration (vertically integrated amount of BC). The contribution of biomass burning to column concentration is very significant and much higher than the surface concentration in spring and especially during spring 2008. The heat and convection caused by the fires inject the biomass burning emissions much higher in the atmosphere; hence the impact of biomass burning emission is accentuated in column concentrations. The biomass burning contribution to Arctic column BC in spring 2008 is almost double that of spring 2009, which shows the impacts of an unusually higher number of forest fires in 2008.

The middle panel of Figure 10 shows the time series of contributions from the emission sectors to anthropogenic $PM_{2.5}$ and biomass burning over the Arctic. Biomass burning contributes to the $PM_{2.5}$ seasonality with maximum contribution in spring and summer. The power, industry, and transportation sectors are the highest contributors during wintertime, reflecting the increased energy consumption for both domestic and industrial heating.

Figure 10- bottom panel shows the contribution of different $PM_{2.5}$ components to the Arctic total $PM_{2.5}$ concentration. BC comprises an average of ~5% of $PM_{2.5}$ over the Arctic. Fine dust (defined as dust with aerodynamic diameter of less than 2.5 µm) is a major source of $PM_{2.5}$ seasonal variation, with maximum contribution in spring (~40%). Sulfate shows the highest contribution over the winter months with a peak of ~60%. Sulfate maximum in winter is caused by the shift in the transport pathways of pollutants during wintertime over the Europe. The high values of the Arctic sulfate during the cold months are partly due to the large Europe contribution with higher use of fossil fuel and coal burning and $SO_2$ emissions for industry, power and residential purposes. The industry and power sectors have the highest contributions to the Arctic sulfate concentration (~43% and ~41%) on an annual basis. The seasonality is described further in section 3.4.

### 3.3.2. Geographical Source Contribution to PM Concentration

Contributions of BC emissions from different source regions (i.e. Europe, China, North America, Central Asia, Middle East, South Asia, Central Asia, and Siberia) were also analyzed through emission perturbation simulations.

Figure **11** shows the spatial plots of annual average contributions of different geographical regions to surface BC concentration with the largest contributions from Europe and China over the Arctic. China also contributes to ~35% of the BC in Canada, Northwestern CONUS and Alaskan regions, which indicates the significance of the inter-continental transport of BC. North American BC emissions have up to ~20% contribution to Southern Europe surface BC concentration.

The source region contributions to surface and column BC concentration exhibit significant seasonal variability. Figure 12 shows the contributions of different emission regions to BC surface concentration and column amounts. Anthropogenic emissions from Europe and China have the highest impact on the Arctic surface BC concentration with annual averages of ~46% and ~25%. However, Europe only contributes to ~25% of the Arctic BC column and China contributes ~36% of column BC in the Arctic. During the winter and spring, air masses from

colder and drier regions can follow surfaces of constant potential temperature and cross the Arctic front barrier but emissions from moister and warmer regions such as North America and China cannot easily cross the Arctic front. However, these particles originating from warmer and moister lower latitudes regions can be lifted and transported to the Arctic in the middle and upper troposphere along the isentropes (AMAP, 2011b; Barrie, 1986; Law and

Stohl, 2007; Stohl, 2006). Therefore, emissions from northern latitudes such as Europe and Russia have higher contributions to the surface concentration but emissions from lower latitudes have higher contributions to the column aerosol load in the Arctic. Anthropogenic emissions from North America (Canada and United States) are also significant contributors to the BC column concentration with contributions of ~10%. However, anthropogenic emissions from North America contribute only ~4% of surface concentration over the Arctic. North American

emissions are mostly from lower latitudes with higher potential temperature and higher humidity. The major transport pathway of North American emissions to the Arctic follows constant potential temperatures, which cause cloud formation and precipitation, hence higher wet scavenging of aerosols. Brock et al., 2011, McConnell, 2007, Stohl, 2006, Breider et al., 2014, and Liu et al., 2015 show similar low contributions of North American anthropogenic emission to the Arctic surface concentration. Less than 5% percent of emissions are transported from

each of South Asia and the Middle East to the Arctic. During the winter, anthropogenic emissions from Russia accounts for ~12% of BC surface concentration and less than 5% of column BC concentration over the Arctic. This is due to the thermally stable condition and lower vertical mixing during the winter over Russia, which facilitates pollution transport to the Arctic. During the spring time, anthropogenic emissions from Europe, China, and Russia account for ~35%, ~25%, and, <~10% of BC surface concentration. This finding is consistent with the study of

Koch and Hansen 2005, which showed that emissions from Russia, Europe and South Asia have contributions of 20-30% during springtime. It should be noted that emission perturbation simulations were not performed on the BC coming from the boundaries and the contributions of BC coming from outside of the modeling domain are not calculated since these emissions are not expected to have a significant impact on the Arctic region. Previous studies have shown that the emissions from regions outside our modeling domain (i.e. Southern America, Australia, Central

and Southern Africa) have insignificant contributions to the Arctic BC burden (Reddy and Boucher, 2007; Wang et al., 2014; Zhang et al., 2015). For example, Reddy and Boucher, 2007 shows that emissions from Australia, South America, and Africa each contribute to ~1% of sum of both Arctic and Antarctic BC surface deposition. Similarly, Wang et al., 2014 study shows the total contributions of emissions in the Southern Hemisphere to the Arctic BC column burden is <1%. Zhang et al., 2015 also indicates that contributions of emissions from Australia

and South America contribute to 0% and <0.1% of the sum of Canada, former Soviet Union and Europe BC burden. Our modeling study shows that the sum of contributions from the Middle East, Central Asia, and South Asia (both

anthropogenic and biomass burning) to the Arctic is ~9%. This is in agreement with AMAP 2015 multi-model study, which lumped the emissions from Middle East, Central Asia, Africa, South Asia, and all emissions from Southern Hemisphere into Rest of the World emission (ROW) emissions (Figure 5-1 of AMAP 2015) and showed that total contributions from ROW (including the regions above) to BC burdens in the Arctic is between ~7% to ~14% (Figure 11-1 of AMAP 2015). Similarly, Ikeda et al., 2017 shows that anthropogenic emissions from regions other than the four primary source regions defined in the study (Europe, East Asia, North America, and Russia) contribute to ~3% of surface BC concentration and ~11% of BC concentration below 5km.

The peak BC surface concentration occurs during the wintertime; however, the contribution of biomass burning in Siberia significantly increases during spring and summer periods, when the biomass burning emissions are the highest. The contributions of Siberian biomass burning to the Arctic surface and column concentration almost doubled during spring 2008 compared to spring 2009. The spring 2008 peak concentrations are explained in the model by Siberian biomass burning plumes transported to the Arctic with low wet scavenging by precipitation and dilution. During the winter, anthropogenic emissions account for ~97% of BC concentration over the Arctic, while during the summer biomass burning contributes up to ~50% of Arctic BC concentration.  During the summer, the contributions of European biomass burning increase. The simulation results also show that the biomass burning plumes from South East Asia can reach the Arctic troposphere accounting for up to ~10% of BC aerosol loading during April 2009.

Figure 13 shows the percentage contributions of various sectors and regions to BC, sulfate, and PM$_{2.5}$ at Alert, Barrow, and the Arctic region average (i.e. 60° N and above). This figure shows that the power and industry are the major sector contributors to sulfate at Alert and Barrow and over the Arctic region, while Europe, China, and Eurasia are the major regional contributors to the Arctic sulfate.  ~~SO$_4$~~ At both Barrow and Alert sites, the contributions of china to surface and column sulfate concentration are at maximum level during summer and fall. The sectoral contributions for sulfate and PM$_{2.5}$ for Barrow are similar to those for the Artic mean. Therefore, Barrow is representative of the sectoral contributions to the Arctic mean sulfate and PM$_{2.5}$. The geographical contributions show more variability between sites and the Arctic mean. However, the geographical contributions to the BC in Alert is a good representation of that of the Arctic average.

For informing more efficient policies, it is essential to study the impact of emissions from various economic sectors of specific source regions on the Arctic surface and column concentrations. Since Europe and China had the highest contributions to the Arctic BC concentrations, the impacts of each specific economic sectors from China

and Europe on the Arctic PM concentrations were studied further. Figure 13 also shows the annual average concentrations of each economic sector from Europe and China to surface BC concentration. The residential sector from China contributes to ~14% of total BC surface concentration over the Arctic. The residential sector accounts for > ~55% of total China contribution to the Arctic surface BC concentration. The emissions originating from residential and transportation sectors in Europe account for ~90% (~55% from residential and ~35% from transportation sectors) of total European contributions to the Arctic surface BC. Figure 12 shows how the contributions of specific emission sectors for China and Europe vary by season. The emissions from European residential sector contributes to ~25% of Arctic BC surface concentration on an annual basis. This impact is much higher during the winter and spring due to higher emissions for heating purposes. Figure 13 (g-l) subplots show the contributions of different economic sectors from China and Europe to the impact of emissions from Europe or China to annual surface BC, sulfate, and $PM_{2.5}$ concentrations for Alert, Barrow, and the Arctic average.

Emissions from Chinese industry sector and European power sector contribute ~12% and ~18% of the Arctic surface sulfate concentration. Emissions from power sector in china also contributes to ~8% of Arctic annual average sulfate concatenation. It should be noted that > 50% and ~35% of China contribution to the Arctic sulfate originated from industry and power sectors respectively (Figure 13). ~80% of Europe contribution to the Arctic sulfate is emitted from power and industry sectors (~45% and ~35%). Emissions originating from power, industry and residential sectors in Europe account for ~12%, ~8%, and ~8% of total $PM_{2.5}$ surface concentration over the Arctic respectively. Further seasonal and spatial analysis (Figure SM10 and SM11) show that Chinese residential emissions have higher impacts (up to ~35%) on the Pacific Arctic (including Siberia, Alaska, Canadian sub-arctic, and Bering Sea) during the winter. Further details on the seasonality of contributions of various emission sectors from Europe or China to the BC surface concentrations over the entire domain are presented in Figure SM10 and Figure SM11.

### 3.4. PM Vertical Profiles and Associated Seasonality

To further understand the seasonal differences in the composition of BC by altitude, the seasonally averaged altitude-latitude cross-sections are shown at 65 °N (entrance boundary for the Arctic) in Figure 14. During the spring, the concentration of BC is relatively high in Eurasia and Siberia. This is partly due to southerly extent of the polar dome during spring especially over Eurasia, which facilitates the transport of BC emission from lower

latitudes to the Arctic. During spring, there are extensive agricultural fires and high number of forest fires in Northern Siberia. In addition, spring 2008 had exceptionally higher numbers (almost double) of Siberian boreal forest fires compared to other years (Liu et al., 2015).

During winter (Figure 14-d), we see higher concentration of BC up to 5km indicating the higher low-level transport of BC from the source regions including North America, Europe and Siberia indicative of stable and low vertical mixing. During the cold months, Europe is the major contributor to the BC concentration, at lower altitudes as shown in Figure 14-l. This is due to thermally stable conditions over winter, which inhibits the upward transport and vertical mixing of emission plumes. However, China shows higher contribution at mid and upper troposphere, which indicates the transport pathways of Asian plumes to the Arctic (Figure 14-h). The contribution of biomass burning to BC concentration is high during summer over Eurasian Arctic, Siberia, and North American Arctic. The contribution of biomass burning is especially high in spring over Siberia during spring 2008 relative to the other years. Also, higher residential emissions of BC in Europe and Asia during the winter is another factor contributing to the higher BC concentration over the Arctic. Siberian forest fires are the major cause of higher BC concentration in Siberian Arctic during summer (Figure 14-n). The higher rate of wet scavenging during summer causes lower transport via low-level pathways. However, the convection caused by forest fires can inject BC in the free troposphere, which reduces the wet and dry deposition for that plume. Figure SM12 shows the dust concentration at the 65 °N cross-section. During spring, we have higher altitude plumes of dust transporting to the Arctic. Dust emission sources are usually from lower latitudes dry and semi-arid regions; hence dust transport to the Arctic is usually higher in the troposphere. Summer also shows similar pattern but with less intensity compared to the spring. Figure SM12 (left column) shows the seasonal and annual sulfate cross-section at 65 °N. For sulfate the cross-sections have similar pattern as BC. However, the concentration of sulfate is much less pronounced during spring compared to BC. During winter high sulfate concentrations were observed under 4km over Eurasia and to the lesser extent Alaska.

## 4. Conclusions and Future Works

In this study, we used a chemical transport model (STEM) to investigate long-range transport of PM to the Arctic and calculate the contributions of various anthropogenic and biomass-burning emission sources to the Arctic surface and column PM concentrations. The focus of this study was to quantify and assess the impacts of different economic source sectors and source regions to the Arctic BC, sulfate ($SO_4$), and $PM_{2.5}$ concentrations using

sensitivity simulations. The simulated BC and sulfate concentrations were evaluated with observations at two Arctic sites (Alert and Barrow). The simulated concentrations were further validated along ARCTAS DC-8 flights and IMPROVE surface sites over the US.

This study found that residential and transportation sector emissions were the major contributors to the Arctic BC loading on an annual basis with contributions of ~38% and ~30% respectively, while power, industrial, and biomass burning emissions were the major contributors to the Arctic $PM_{2.5}$ (contributions of ~30%, ~25%, and ~20% respectively). The simulations showed a distinct seasonality in the contributions of economic sectors and source regions to BC and $PM_{2.5}$ concentration over the Arctic. During the winter peak concentration period, the contributions from residential sector were highest due to high-energy consumption for heating purposes. Biomass burning also showed a distinct cycle with contributions to BC surface concertation as high as ~50% during summer and less than ~3% during winter. The contributions of anthropogenic sources to BC concentrations near the surface were dominant varying from ~50% in spring to ~97% in winter. However, the contributions of biomass burning from Siberia were significant during spring 2008 (up to ~40%), and the contributions of biomass burning emissions from Europe became significant over the summer accounting for up to ~20% of Arctic BC column concentration. There is also a distinct inter-annual difference in BC from biomass burning between spring 2008 and spring 2009 which indicates the higher occurrence of fire during spring 2008. Biomass burning plumes from South East Asia can reach the Arctic troposphere accounting for up to ~10% of BC column concentration during April 2009.

Industrial and power emissions had the highest contributions to the Arctic sulfate surface concentration with annual contributions of ~43% and ~41% respectively. The dominant source region for the Arctic sulfate surface concentration is China, Europe, and Russia. Emissions from power sector in Europe and industry sector in china contributes to ~ 18% and ~12% of Arctic sulfate concentration.

Fine dust was shown to be one of the most important drivers of Arctic $PM_{2.5}$ seasonality, with maximum contributions in spring (~40%). Dust was the largest component of $PM_{2.5}$ in the region in all seasons except for cold months, when sulfate was the largest contributor (~60%) to the $PM_{2.5}$.

In this study, we found that the major source regions contributing to BC surface concentrations are Europe and China annually with contributions ~46% and ~25% respectively. Among the various economic sectors from each of the geographic regions, the residential sector from Europe and China were the largest contributors to Arctic BC with ~25% and ~14% respectively. In addition, the contribution of each geographic source region varied significantly by altitude. In the mid and upper troposphere, the contributions of Chinese emissions were higher due

to their dominant transport pathway to the Arctic though warm conveyer belts. Model results showed a distinctive temporal variability for regional contributions to the Arctic. In general, the anthropogenic emissions from Europe were the most significant due to its large contributions over the winter (haze season).

There are a number of factors (including high uncertainties in emission inventories, transport pathways, and removal parametrizations) that can contribute to uncertainties associated with the contributions of individual source sector and source regions to the Arctic PM loading. Future Arctic warming, sea ice decline, and industrial development facilitate international shipping and transport via the northern sea route, which consequently increase the Arctic pollutants burden (Law et al., 2017; Marelle et al., 2016). Additional observations at Arctic locations along with higher resolution and more sophisticated modeling studies are necessary to reduce these uncertainties in future. Improved estimates of local Arctic emissions are essential for developing successful pollution mitigation strategies.

## Acknowledgements

The University of Iowa group activities are funded by NASA awards (NNX08H56G and NNX12AB78G) and an EPA award (RD-83503701-0). The authors would like to acknowledge ARCTAS science team. We would also like to thank the Global Monitoring Division (GMD) at NOAA Earth System Research Laboratory (ESRL), and Environment and Climate Change Canada for maintaining and providing BC measurements. We would like to acknowledge World Meteorological Organization (WMO) Global Atmosphere Watch (GAW) and the Norwegian Institute for Air Research (NILU) for hosting the measurement data (http://ebas.nilu.no). We also would like to thank the Emissions Database for Global Atmospheric Research (EDGAR) for developing HTAP emission inventories. The authors would like to thank Christine Wiedinmyer for providing open burning emission data. The authors would like to thank Interagency Monitoring of Protected Visual Environments (IMPROVE) for providing observational data over the CONUS. IMPROVE is a collaborative association of state, tribal, and federal agencies, and international partners. US Environmental Protection Agency is the primary funding source, with contracting and research support from the National Park Service. The Air Quality Group at the University of California, Davis is the central analytical laboratory, with ion analysis provided by Research Triangle Institute, and carbon analysis provided by Desert Research Institute. The National Center for Atmospheric Research (NCAR) is sponsored by the National Science Foundation (NSF).

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

# 7. Tables and Figures

**Table 1: NASA ARCTAS Flight Categories for Spring and Summer 2008**

| Flight Season | Flight Categories | Flight Date | Flight Number |
|---|---|---|---|
| Spring Flights | Spring Alaska Local Flights | 04/12/2008 | 08 |
| | | 04/16/2008 | 09 |
| | Spring Greenland Flights | 04/04/2008 | 04 |
| | | 04/05/2008 | 05 |
| | | 04/08/2008 | 06 |
| | | 04/09/2008 | 07 |
| | | 04/17/2008 | 10 |
| | Spring Transit Flights | 04/01/2008 | 03 |
| | | 04/19/2008 | 11 |
| Summer Flights | Summer California Flights | 06/18/2008 | 12 |
| | | 06/20/2008 | 13 |
| | | 06/22/2008 | 14 |
| | | 06/24/2008 | 15 |
| | Summer Canada Flights | 06/29/2008 | 17 |
| | | 07/01/2008 | 18 |
| | | 07/04/2008 | 19 |
| | | 07/05/2008 | 20 |
| | Summer Canada Greenland Flights | 07/08/2008 | 21 |
| | | 07/09/2008 | 22 |
| | | 07/10/2008 | 23 |
| | Summer Transit Flights | 06/26/2008 | 16 |
| | | 07/13/2008 | 24 |

**Table 2- Statistical summary of comparison of observed and modeled meteorological parameters for NASA ARCTAS spring and summer flights. Obs and Mdl denote observation and model data.**

| | Temperature (K) | | Pressure (hpa) | | Relative Humidity (%) | | Wind Speed (m/s) | |
|---|---|---|---|---|---|---|---|---|
| | Obs | Mdl | Obs | Mdl | Obs | Mdl | Obs | Mdl |
| Mean | 248.4 | 263.1 | 610.2 | 594.6 | 45.5 | 45.5 | 13.0 | 13.5 |
| Standard Error | 0.3 | 0.3 | 3.7 | 3.4 | 0.4 | 0.4 | 0.2 | 0.1 |
| Median | 245.4 | 265.7 | 554.9 | 569.0 | 43.4 | 43.4 | 9.7 | 11.5 |
| Mode | 225.0 | 231.4 | 1007.0 | 329.3 | 19.8 | 19.8 | 25.7 | 25.7 |
| Standard Deviation | 23.6 | 23.1 | 253.2 | 232.8 | 27.0 | 27.0 | 10.9 | 9.1 |
| Range | 94.8 | 93.6 | 818.9 | 817.2 | 117.1 | 117.1 | 56.2 | 43.4 |
| Minimum | 212.7 | 212.2 | 206.7 | 187.1 | 0.7 | 0.7 | 0.2 | 0.1 |
| Maximum | 307.4 | 305.8 | 1025.6 | 1004.2 | 117.8 | 117.8 | 56.4 | 43.5 |
| R-Square | 0.984 | | 0.757 | | 0.585 | | 0.405 | |
| Standard Error | 32.463 | | 314.263 | | 34.059 | | 12.553 | |

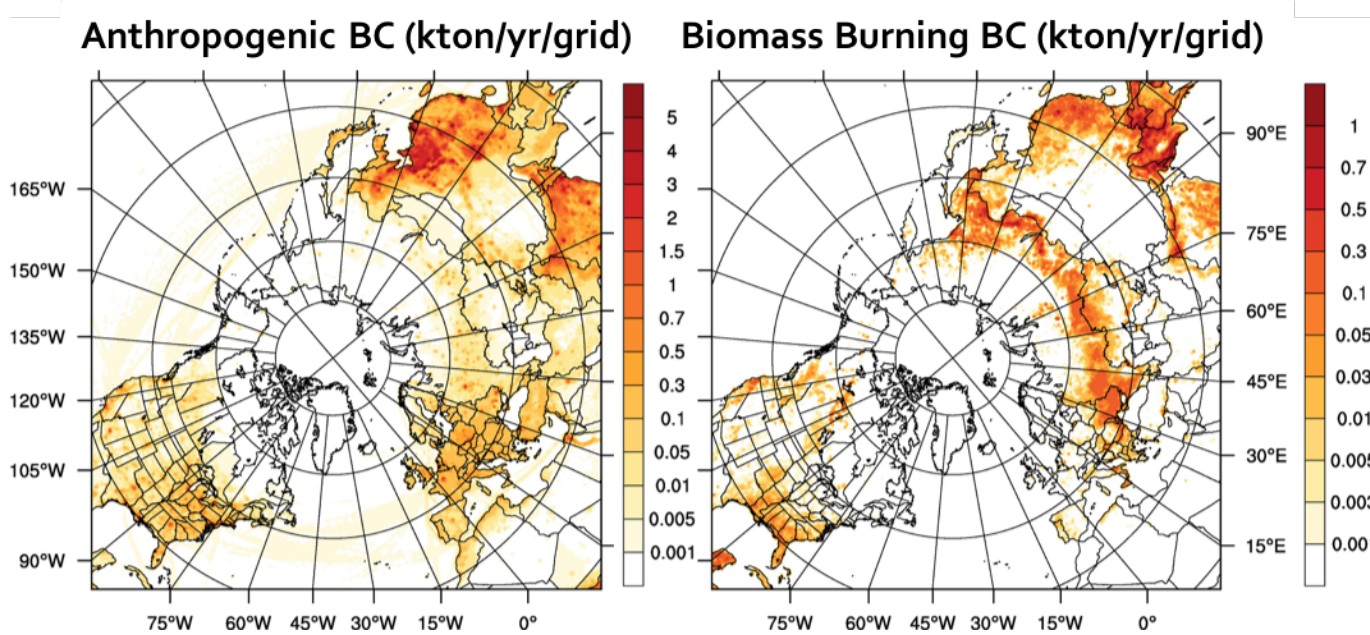

**Figure 1: Spatial distribution of anthropogenic BC emissions (left) and wildfire BC emissions (right) in Gg/yr/grid. This figure is generated using NCAR Command Language (NCL) version 6.3.0, open source software free to public, by UCAR/NCAR/CISL/TDD, http://dx.doi.org/10.5065/D6WD3XH5.**

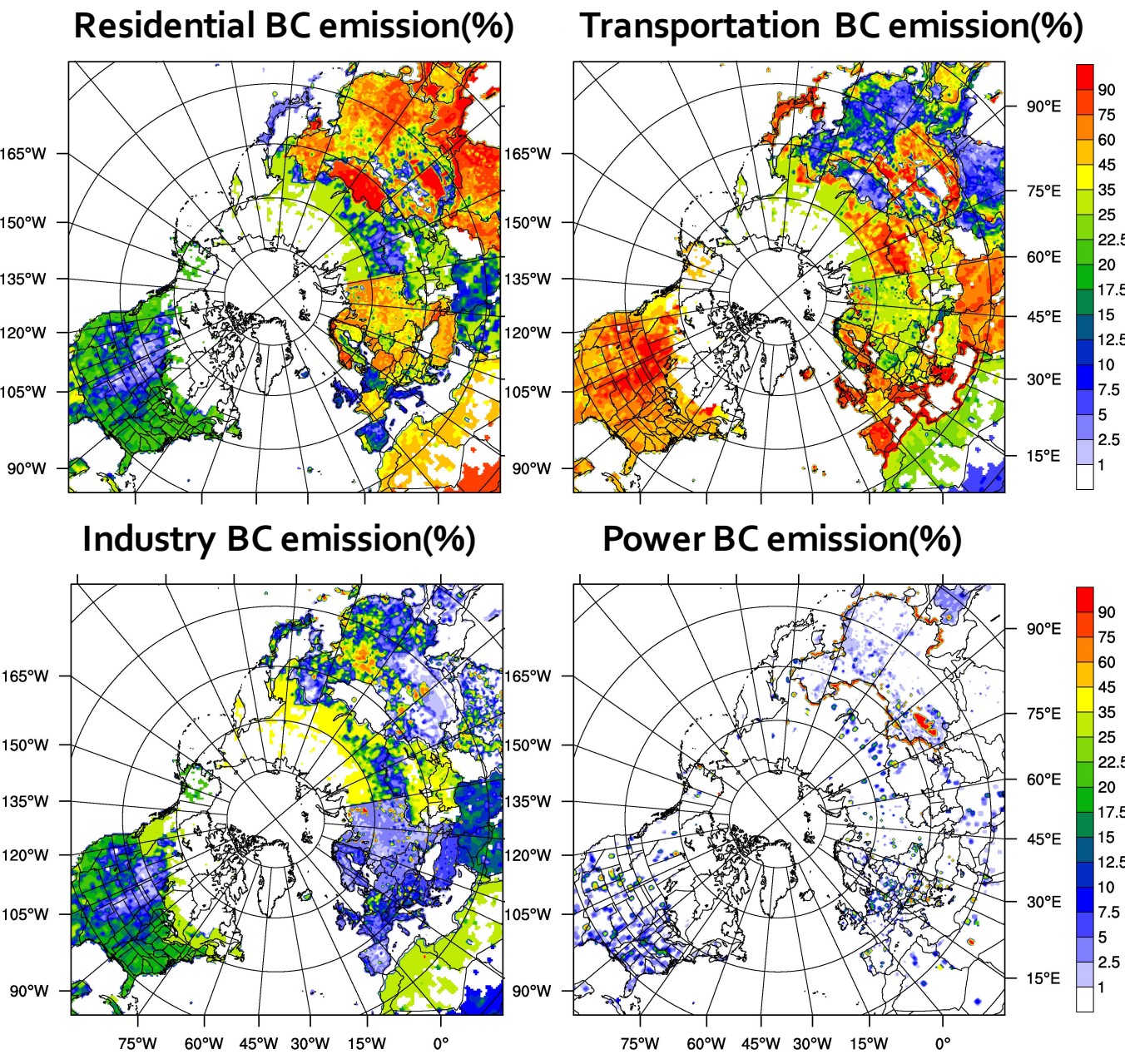

**Figure 2: Spatial distribution of economic sectors (%) to total BC anthropogenic emissions on an annual basis. This figure is generated using NCAR Command Language (NCL) version 6.3.0, open source software free to public, by UCAR/NCAR/CISL/TDD, http://dx.doi.org/10.5065/D6WD3XH5.**

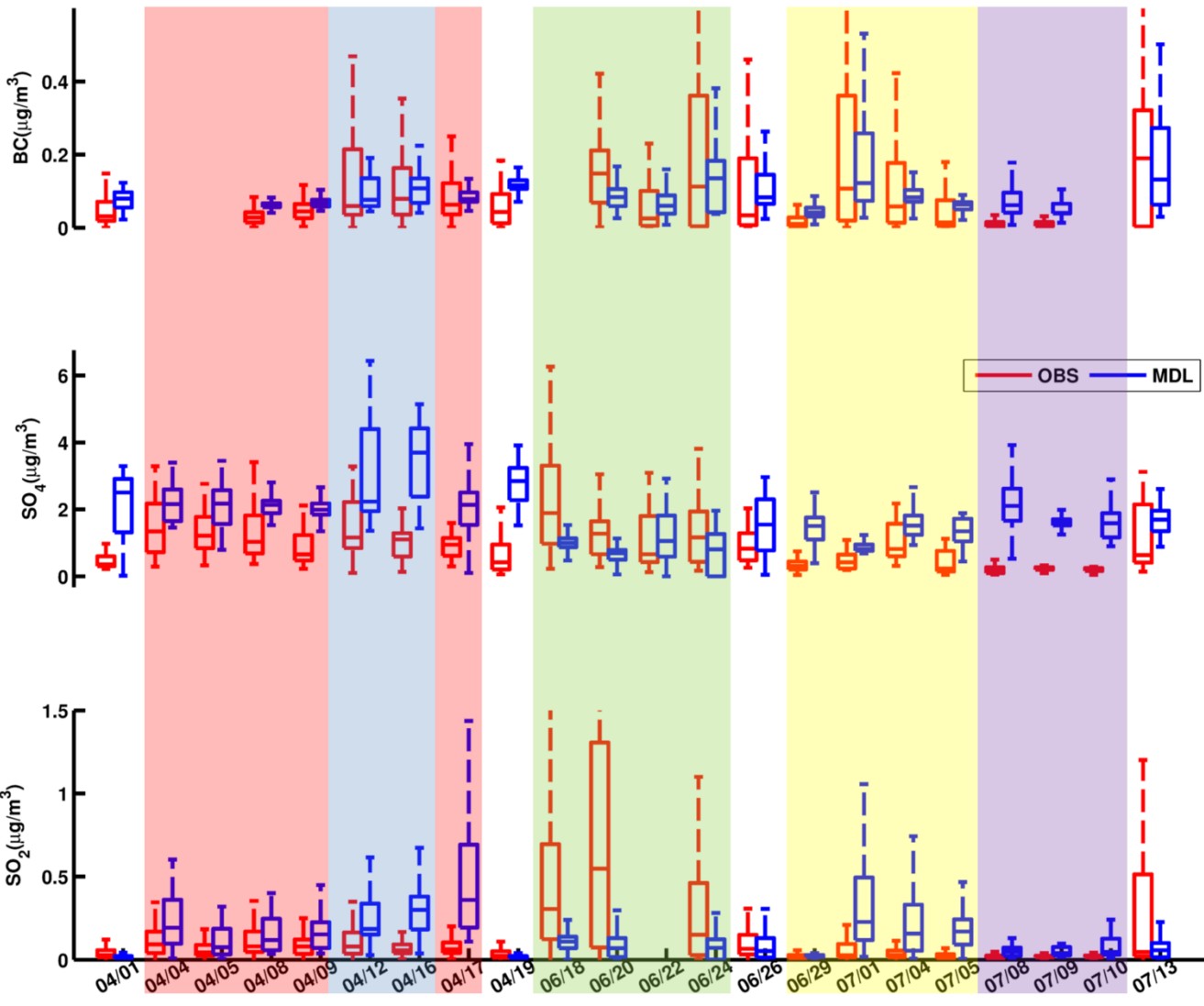

**Figure 3: Comparison of BC, sulfate (SO₄), and SO₂ for NASA ARCTAS spring and summer flights. Flight categories are shaded same as figure 3.  Each flight category is shaded with a different color and the spring and summer transition flights are not shaded. Spring Alaska local flights and spring Greenland flights are shaded blue and red respectively. Green, yellow, and purple shades denote the summer California flights, summer Canada flights, and summer Canada Greenland flights. In each box whisker panel, the middle line denotes the median value, while the edges of the box represent 25th and 75th percentile values respectively. The whiskers denote the maximum and minimum values.**

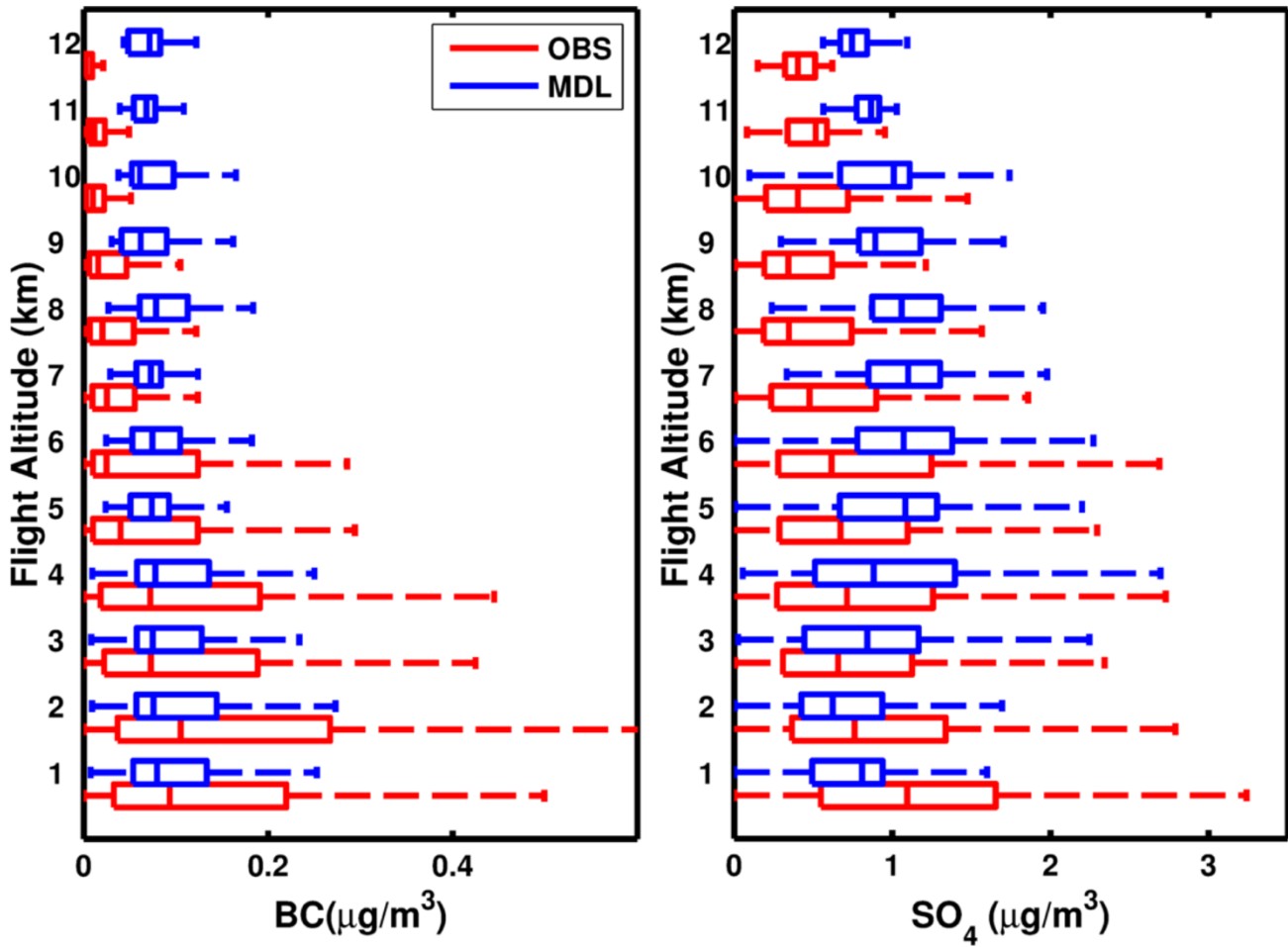

**Figure 4: Comparison of model and observation BC and sulfate (SO₄) for all ARCTAS flights. In each box whisker panel, the middle line denotes the median value, while the edges of the box represent 25th and 75th percentile values respectively. The whiskers denote the maximum and minimum values.**

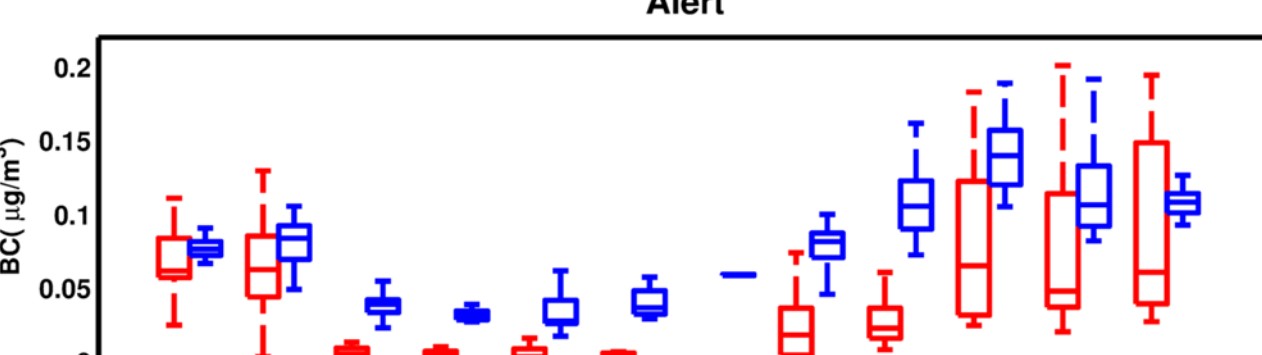

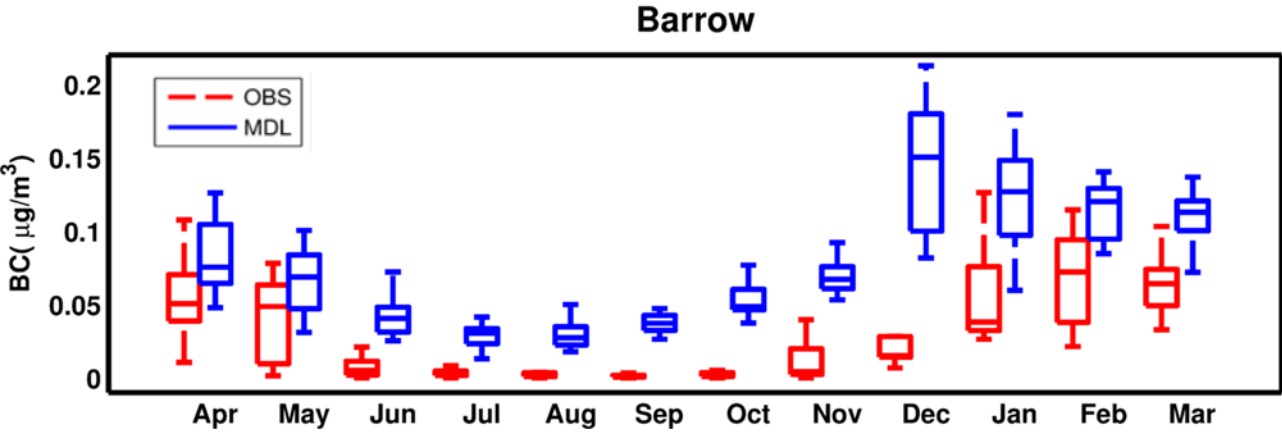

**Figure 5: Comparison of simulated BC with observations shown as box-and-whisker plots over the simulations period at Alert (top panel) and Barrow (bottom panel) sites. In box and whisker, the middle line denotes the median value, while the edges of the box represent 25th and 75th percentile values respectively. The whiskers denote the maximum and minimum values.**

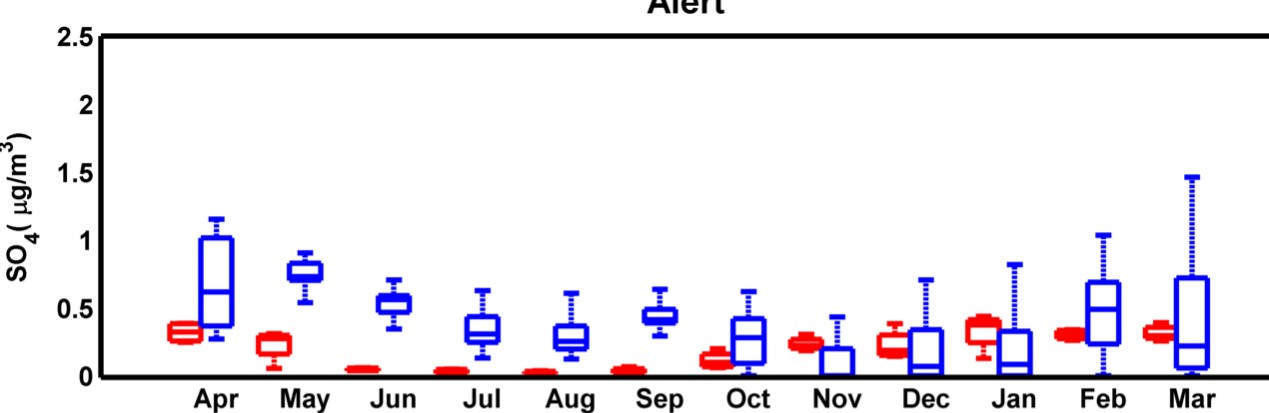

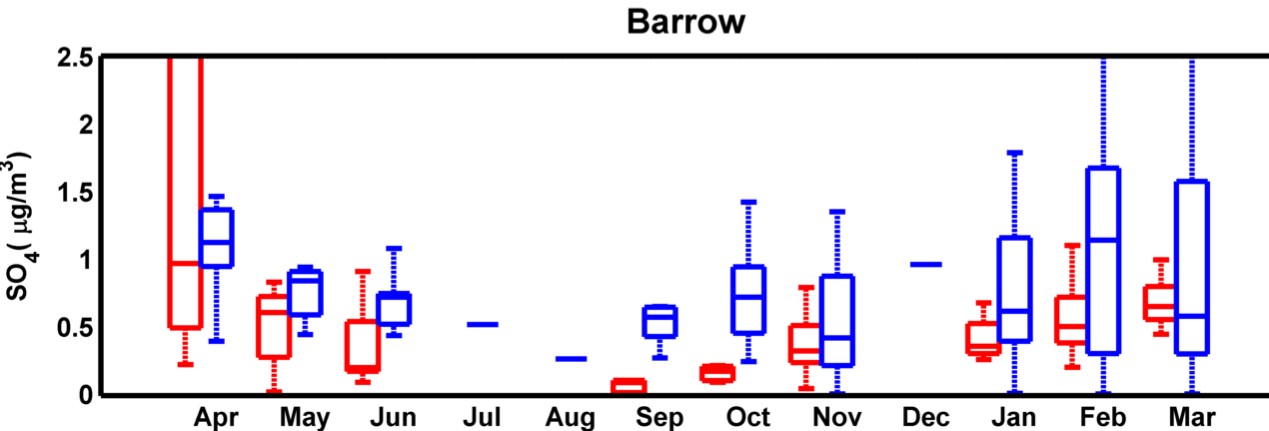

**Figure 6: Comparison of simulated sulfate (SO₄) with observations shown as box-and-whisker plots over one year (April 2008- Mar 2009) at Alert (top panel) and Barrow (bottom panel) sites. In box and whisker, the middle line denotes the median value, while the edges of the box represent 25th and 75th percentile values respectively. The whiskers denote the maximum and minimum values.**

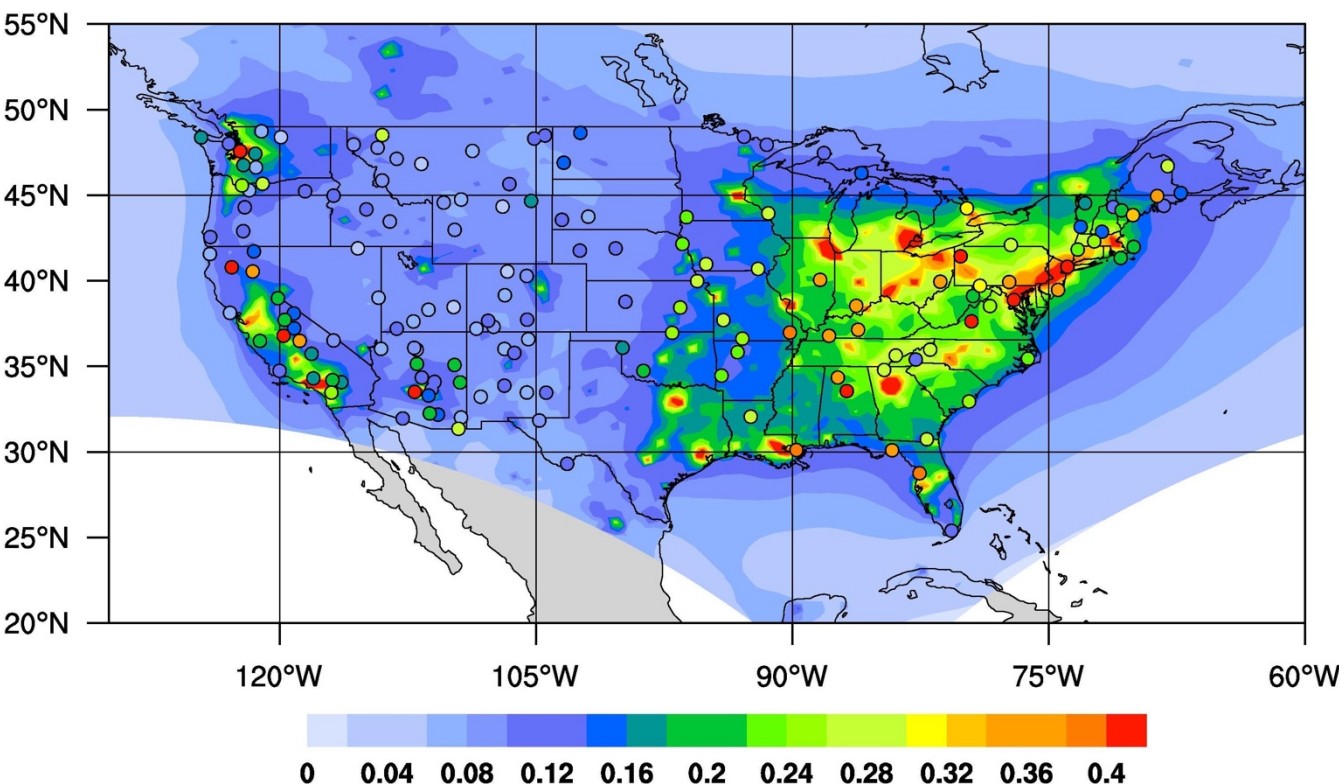

**Figure 7: Annual average surface BC concentration over the U.S. The simulated BC concentration (solid contours) for April 2008 to Mar 2009 are compared to observations (circles) from IMPROVE network. The circles indicate IMPROVE sites with the color representing the BC concentration in μg/m³. This figure is generated using NCAR Command Language (NCL) version 6.3.0, open source software free to public, by UCAR/NCAR/CISL/TDD, http://dx.doi.org/10.5065/D6WD3XH5.**

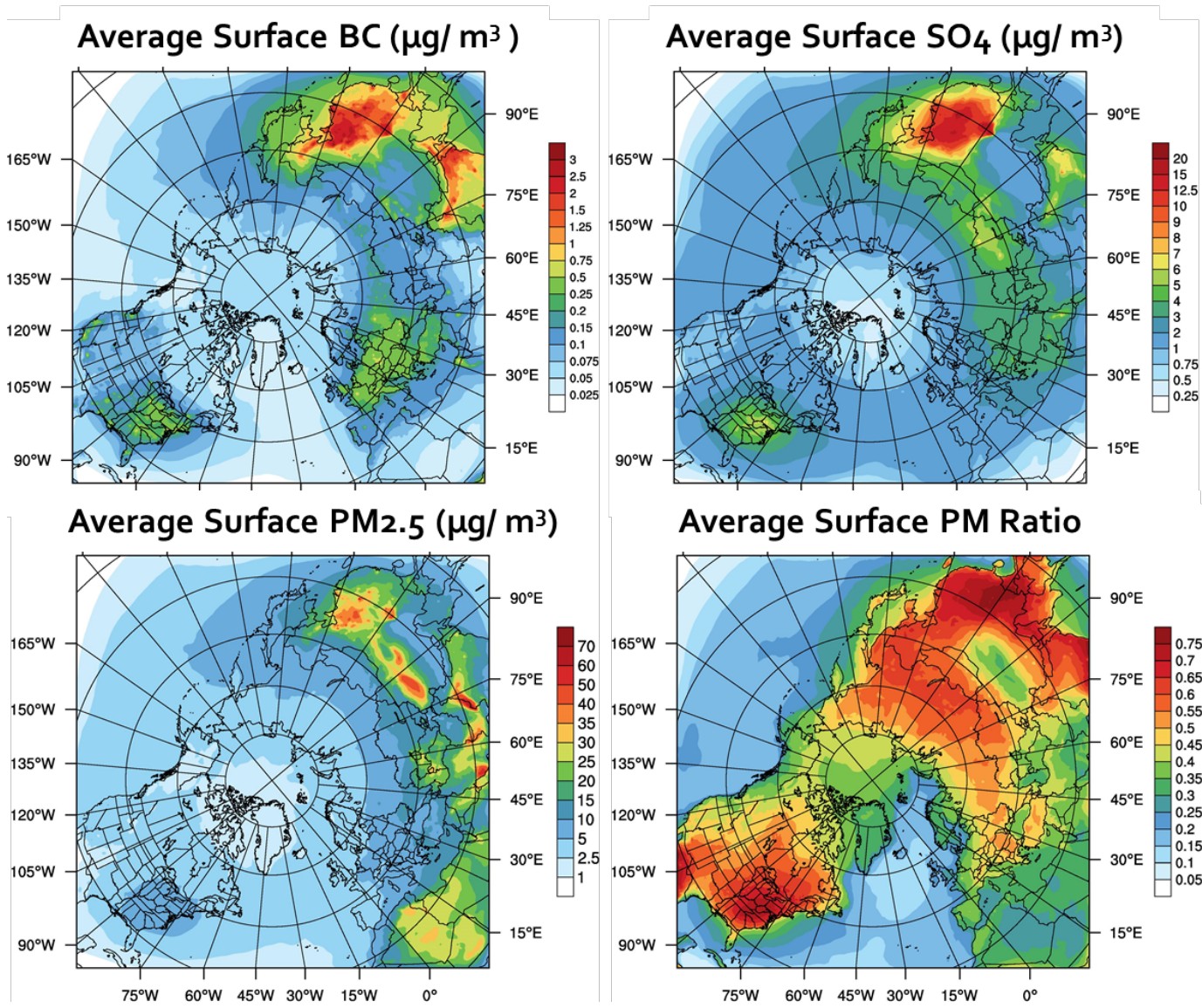

**Figure 8: Spatial distributions of simulated BC (μg/m3), sulfate ($SO_4$) (μg/m$^3$),** $PM_{2.5}$ **(μg/m$^3$), and** $PM_{2.5}$/$PM_{10}$ **ratio averaged over the simulation period. The annual averages are calculated by averaging model outputs from April, 01 2008 to March, 31, 2009. This figure is generated using NCAR Command Language (NCL) version 6.3.0, open source software free to public, by UCAR/NCAR/CISL/TDD, http://dx.doi.org/10.5065/D6WD3XH5.**

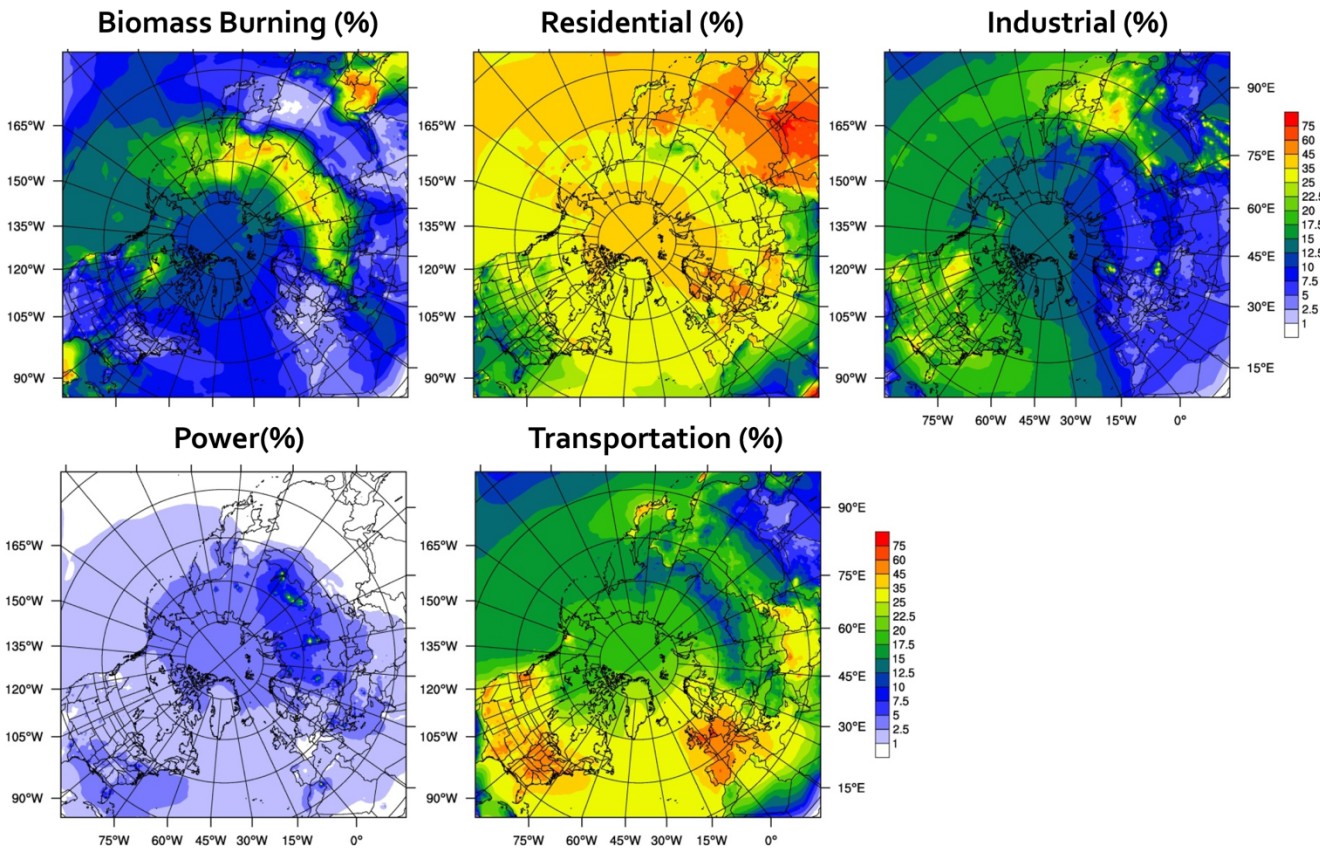

**Figure 9: Spatial distribution of source sector contributions (%) to annual BC surface concentration over the entire domain. This figure is generated using NCAR Command Language (NCL) version 6.3.0, open source software free to public, by UCAR/NCAR/CISL/TDD, http://dx.doi.org/10.5065/D6WD3XH5.**

**Arctic Surface Concentration(μg/ m³ )**

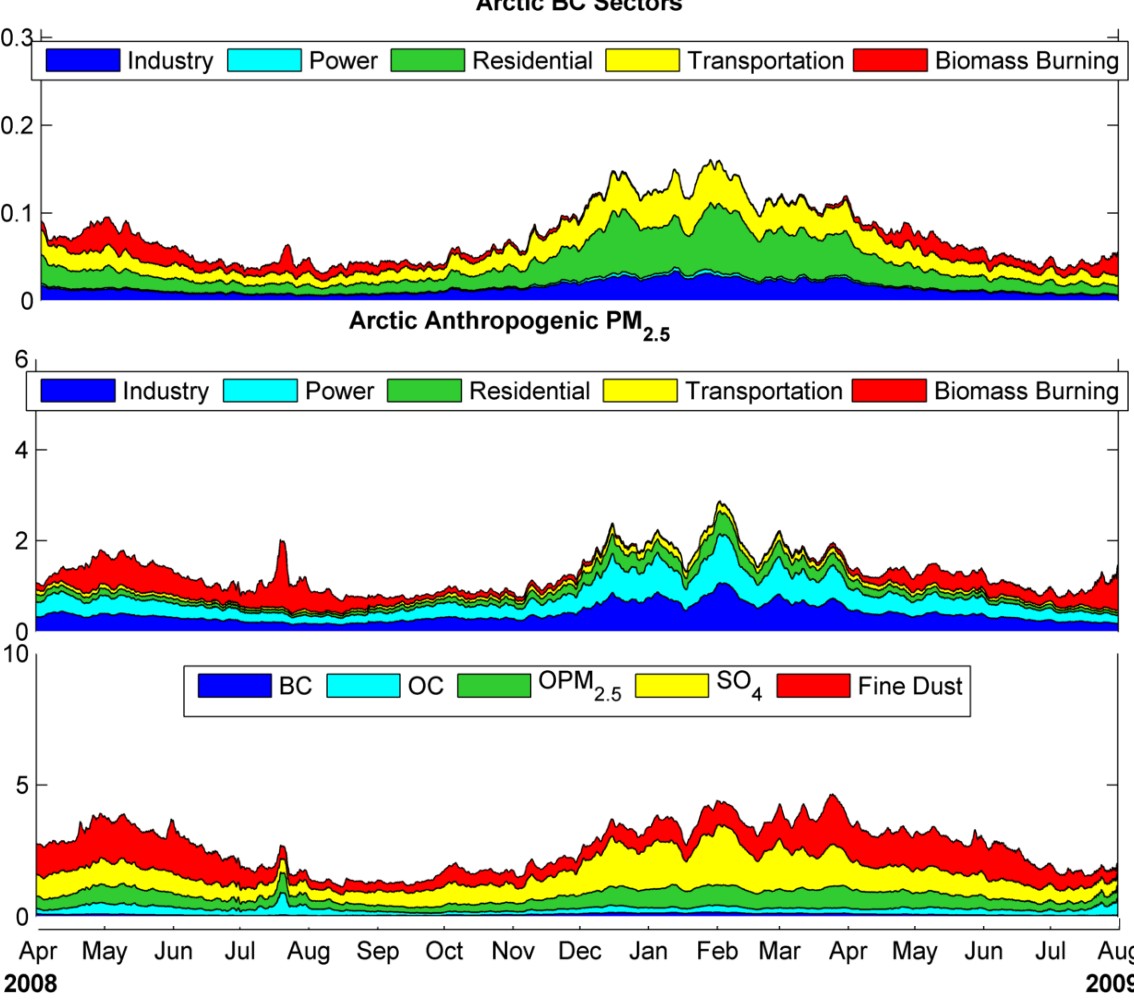

**Figure 10: Time-series concentration and contribution of different sector to BC concentration (top panel) different sectors to PM₂.₅ concentration (middle panel), and different PM₂.₅ species (bottom panel). OPM₂.₅ is the acronym for other PM₂.₅ and refers to other primary emitted non-carbonaceous particles with aerodynamic diameters less than 2.5 μm such as fly ash, road dust, and cement which were simulated as a single mass component in the model.**

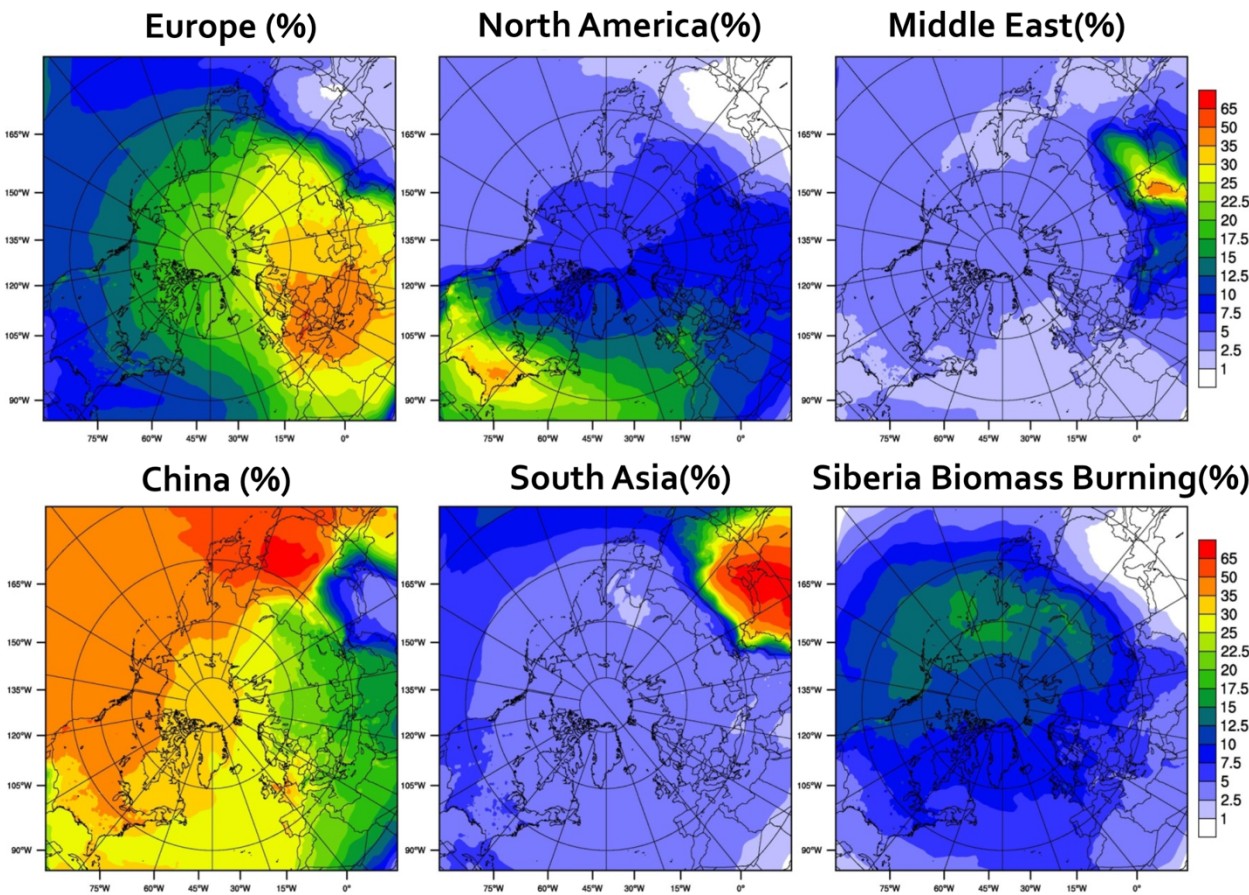

**Figure 11: Spatial distribution of source region contributions (%) to annual BC surface concentration over the entire domain. This figure is generated using NCAR Command Language (NCL) version 6.3.0, open source software free to public, by UCAR/NCAR/CISL/TDD, http://dx.doi.org/10.5065/D6WD3XH5.**

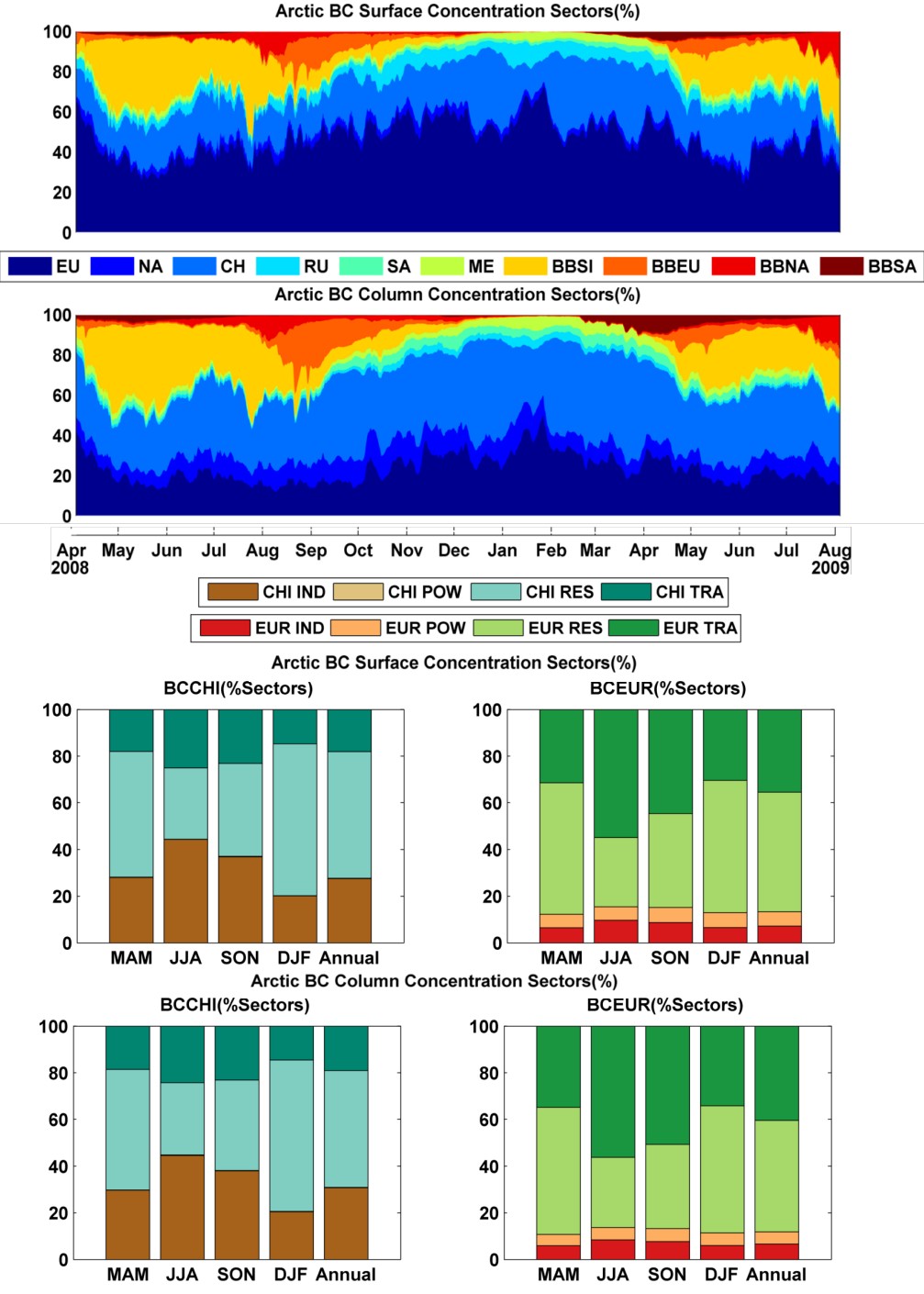

**Figure 12: Top two subplots show the seasonality of BC major geographical contributors to the Arctic (latitudes > 60 °N)surface (first row) and column (second row) concentrations. The bar plots (bottom 4 subplots) indicate the seasonality and annual average of contributions of various economic sectors from**

**Europe or China to the Arctic (latitudes > 60 °N ) surface (third row) or column (bottom row) BC concentration. BBSI, BBEU, BBNA, and BBSA denote biomass burning from Russia, Europe, North America, and South Asia respectively. EUR and CHI denote Europe and China. Industry, power, residential and transportation sectors are represented with IND, POW, RES, and TRA acronyms. MAM denotes the average for months of March, April, and May. JJA denotes the average for months of June, July, and August. SON (bottom right panel) denotes average for months of September, October, and November. DJF denotes the average for the months of December, January, and February.**

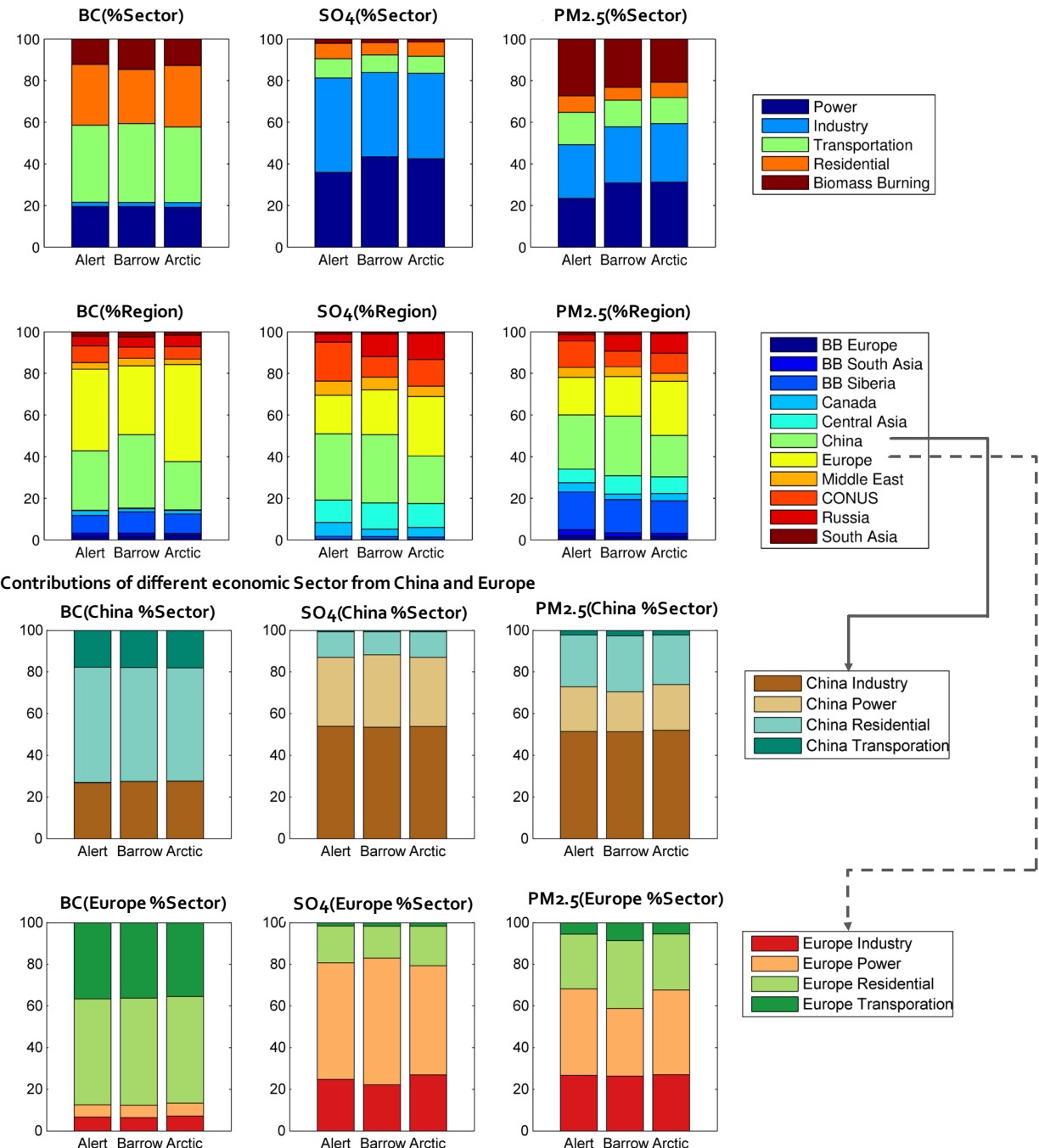

**Figure 13: Summary of annual mean contributions to BC, sulfate (SO₄), and PM₂.₅ by source sectors (top row) , and source regions (2nd row) at Alert, Barrow, and over the Arctic regions. The bottom two rows of bar plots show the relative contributions of various economic sectors from either China (3rd row) or Europe (bottom row) to total China or Europe contributions to Arctic BC, sulfate (SO₄) , and PM₂.₅ concentration. BB denote biomass burning in this figure.**

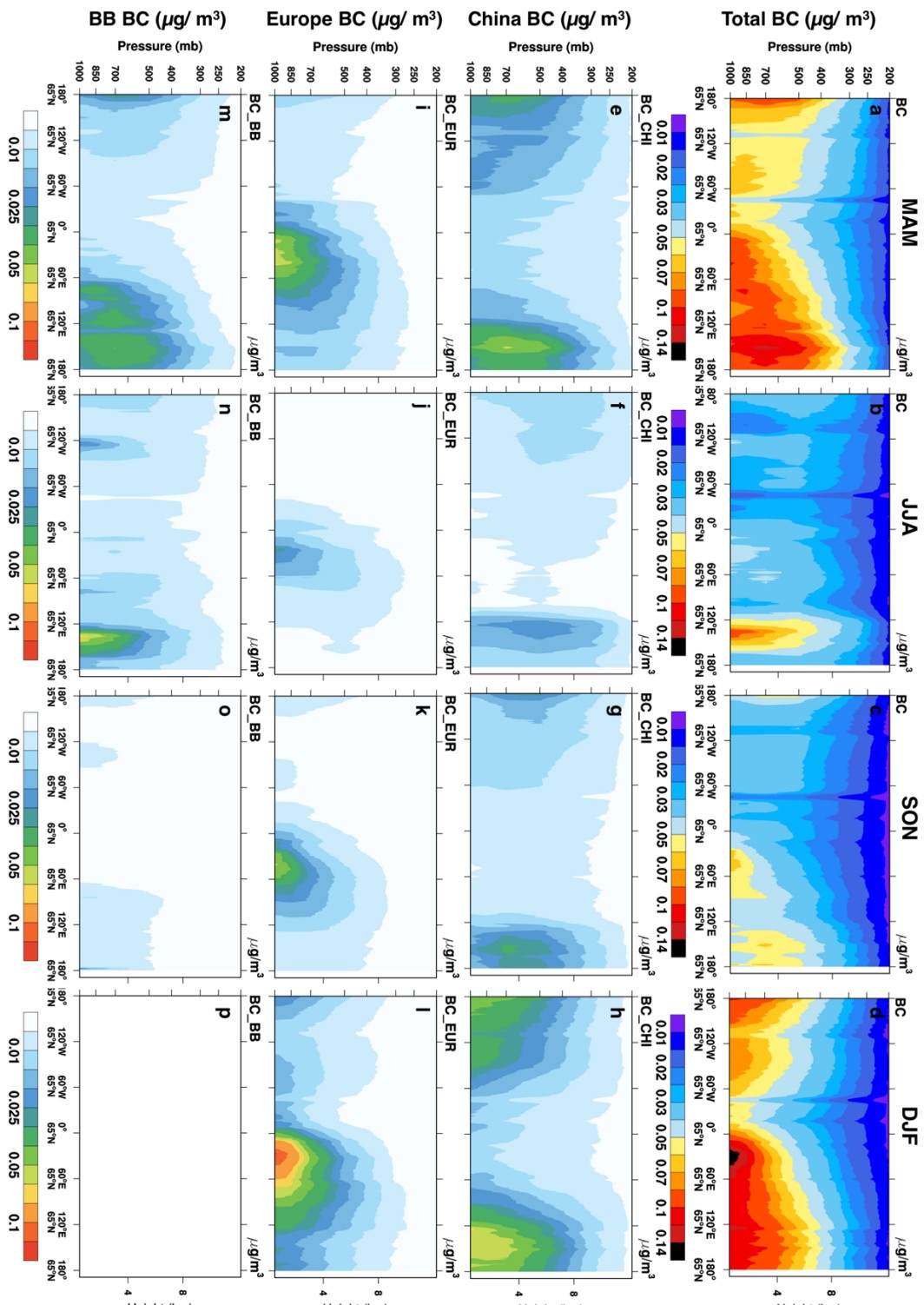

**Figure 14: Cross-section at 65 °N for different seasons. The top row shows the BC concentrations (in µg/m3) at the 65 °N cross-section. The 2nd, 3rd and 4th rows show the contributions of China, Europe and biomass burning (BB) to BC at 65 °N. JJA denotes the average for months of June, July, and August. SON (bottom right panel) denotes average for months of September, October, and November. DJF denotes the average for the months of December, January, and February. This figure is generated using NCAR Command Language (NCL) version 6.3.0, open source software free to public, by UCAR/NCAR/CISL/TDD, http://dx.doi.org/10.5065/D6WD3XH5.**

