# Peer review of "Source Sector and Region Contributions to Black Carbon and PM2.5 in the Arctic"

_Atmospheric Chemistry and Physics, 2018_

## Referee Comment (RC1) · Anonymous Referee #1 · 14 Mar 2018

This study quantifies the contributions of different emission regions and sectors to Arctic Black Carbon (BC) and PM2.5 concentrations using a chemistry transport model. First, the authors evaluate their model to the ARCTAS flight campaign and two stations in the Arctic. Then, by using a sensitivity analysis they identify the main sectors contributing to Arctic PM2.5 (the power, industrial and biomass burning sector), and the largest contributors to BC surface concentrations (the residual and transport sector). Further, they look at the seasonal cycle and emissions from Europe and China in particular.

The study is well-written and concise and the figures are clear and easy to understand. I really appreciate the seasonal focus in this study. I recommend this manuscript for publication after some clarifications, more details given below.

[Figure]

1. Methods:

- Emissions. Can you please add a paragraph describing the emission dataset in more detail instead of just referring to Kulkarni et al. 2015? E.g. can you explain the different sectors? I like the plots of the data showing the geographical distribution (Fig 1), but could you also say something about the seasonal cycle? Do all sectors have a seasonal cycle? How is this calculated? E.g. is there a correlation with outdoor temperature in the residual sector? Is the seasonal cycle the same every year except for biomass burning? Could you say something about the uncertainties in the emissions?

- STEM model coupled to WRF. Could you also add a paragraph describing this model? Are there any known biases?

- How is the model setup regarding the sensitivity analysis? How did you perform the experiments? Simulation period? Please add a paragraph describing this as well.

2. Figures:

In general, I think the captions could contain more information about the data shown (and what the boxes, lines etc. represent).

FIG3: I suggest moving figure 3 to the Supplementary.

FIG4: Could you add more text in the manuscript on Figure 4? What does the boxes represent? Is it each flight averaged over the column? Maybe you could replace this figure with SM3, which is easier to read I think. Or is your point here to show the flight by flight variation?

FIG5: Again, can you explain what the boxes and whiskers represent? The model, as you say, is biased high at higher altitudes. This is common for many models. Could you maybe speculate a little on the reasons for why your model overestimate BC at high altitude, as you do for SO4?

FIG6: Impressive seasonal cycle in the model. Not many models have such a distinct

seasonal cycle. To be clear: in the data at Barrow used here the BB contributions were removed, right?

FIG7: Could you make the dots representing the IMPROVE observations larger? It is a bit hard to spot them. In the text you say that this is 'annual mean', but the simulation period is April 2008 to July 2009. How did you average the data? Over the whole period? Is 0.16 ug/m3 averaged over the entire US in the model? If so, is that number very different by just averaging the grid boxes containing stations? Are the IMPROVE monthly or daily data? How did you compare missing data etc?

FIG8: The caption says 'Dust' for the 'SO4' plot. Again, you say in the text that this is annual average, but in the caption that this is an average over the simulation period?

FIG10: OPM2.5? This is the first time we see OC, and it is not mentioned in the text.

FIG14: The row with the Dust plot seems a bit out-of-place here. At least change the 'Base' to BC in the first row and have the same color scale as dust? Is this fine-mode dust as in FIG10?

3. Conclusions:

Last paragraph: Could you be a bit more specific here? I miss a discussion on the uncertainties in the emission inventories, observations and the data you have used, but that can be written elsewhere e.g. in the Methods section. You highlight high-resolution modeling; why would high resolution modeling studies be the most necessary to reduce the uncertainties in the future? Do your conclusions fit well with what other studies have shown? What are the implications of your study?

Minor:

Page 2, L8: With 'albedo reduction' do you mean the general warmer temperatures → sea-ice is melting → more open water, or the albedo reduction caused by BC deposition only? If the latter, as it is written now it seems like the largest feedback in the Arctic causing the doubling of temperature increase is caused by BC.

Page 4: L1: You say here that you studied organic carbon? As far as I can see, there is no mentioning of OC except that it is included in the bottom panel of FIG10. And what about dust?

Page 9, L21-23: Where is this shown? Can you point us to a figure?

Page 9, L24: What is the average in the Arctic region? You compare your range to Sharma et al 2013; can you add a sentence about what this number (0.06 ug/m3) represents (e.g. all surface stations?)?

Page 9, L28: Can you add chapter reference for IPCC instead of just the whole report?

Page 10, L10: Can you add why ($\sim$ the natural sources are larger in size)?

Page 10, L18: 'global warming' or you only mean (local) warming over the Arctic?

Page 11, L1: Could you add numbers?

Page 12, L25. Move or remove the parenthesis

Page 15: L4: I guess you refer to 14-l (not 14-i)

---

## Referee Comment (RC2) · Anonymous Referee #2 · 16 Mar 2018

General comments:

The study adds to a growing list of studies on impacts of regional emissions of black carbon on Arctic aerosol concentrations. Given large yet uncertain impacts of aerosols on Arctic climate, there is considerable need for research in this area.

The authors of the study analysed at a wide range of model results in great detail. However, there are substantial issues with the design of the study and presentation of the results.

First, important sources of black carbon were not accounted for. Consequently, BC concentrations in the Arctic are underestimated and conclusions about the relative importance of different source regions are biased. More specifically, emissions of black

carbon from central and south America, central and south Africa, and Australia are not accounted for in the simulations. According to data sets that were used by AMAP for an assessment of the impacts of black carbon on the Arctic (AMAP, 2015; Stohl et al., 2015), these regions contributed about 41% to total (anthropogenic and natural) global emissions of black carbon in 2010 (AMAP region "ROW"). The simulated contribution of these emissions to total black carbon mass in the Arctic atmosphere is 10-20%, depending on the model. Furthermore, it appears that emissions associated with oil and gas flaring were not accounted for. According to AMAP data, about 65 GgC of black carbon were emitted by oil and gas flaring industrial activities in 2010, especially from sources at high latitudes. Efficient transport to the Arctic implies that the contribution of these emissions to total black carbon mass in the Arctic atmosphere is relatively large, i.e. comparable to the impacts of emissions of black carbon from all North American sources.

Second, the analysis of model results in the study seems narrowly focussed on results for the Arctic, which is problematic with regard to an improved understanding of the impacts of aerosols on Arctic climate. For instance, aerosol radiative effects at mid latitudes have a strong impact on Arctic climate by influencing the transport of heat to the Arctic. There are also impacts of black carbon on snow albedos, which are not considered either.

Finally, it is not obvious why results for PM2.5, PM10, sulphate, and dust concentrations are analysed. Comparisons with observations are missing and analysis of model results for these is less complete than the analysis for black carbon. Overall, the relevance of model results is not obvious with regard to impacts of aerosols on climate or air quality. There is no discussion of climate implications or comparison with results from other studies.

Specific comments:

P. 2, L. 22: Another great reference is Bond et al. (2013).

P. 3, L. 10: Sharma et al. (2013) and references given in that paper seem relevant here, too.

P. 3, L. 27: Uncertainties and biases in parameterizations of wet deposition and convective processes should be more emphasized here since they mainly explain differences in simulated aerosol concentrations in the Arctic. See Browse et al. (2012) and Mahmood et al. (2016) and references in these papers.

P. 3, L. 28: The statement that regional chemical transport models capture BC concentrations better is highly questionable. According to Eckhardt et al. (2015), there is no single class of models that outperforms other models. The statement here is based on 2 individual models.

P. 3: The discussion in this section is focussed on simulations of surface concentrations. For climate and radiative forcings, aerosol vertical distributions and deposition of BC on snow are more important than surface concentrations. Please clarify the focus of the study on air quality aspects of Arctic aerosols or include further information regarding aerosol vertical profiles. This is particularly important since comparisons for aerosol vertical profiles are included in the paper (Section 3.1.2). The title and abstract are somewhat ambiguous in that regard.

P. 4, Section 2.1.2: Please indicate whether annual mean emissions are used. Are temporal variations in fire emissions are accounted for?

P. 5, Section 2.1.3: This section lacks a lot of detail. What types of aerosols are consider and how they are numerically represented? Does the model simulate aerosol size distributions? What kind of aerosol processes are accounted for? Is aging of BC accounted for? How are interactions between aerosols and gas-phase chemistry represented?

P. 7, Section 3.1.2: Comparisons for BC are included here but similar comparisons for sulfate are missing. The latter needs to be added (e.g. by comparing sulfate concentrations in the Arctic with observations) because much of the subsequent discussion in the manuscript also addresses sulfate. It is not clear how well the model simulates non-BC aerosol species. Similar, it is not obvious that simulated results for PM are realistic. Without validation it would be better to remove results for sulfate and PM from the paper altogether.

P. 11, L. 23: Figure 10 shows the seasonal cycle of aerosol concentrations at the surface. A similar figure needs to be added for the vertically integrated amount of BC in the Arctic. This information is necessary for the interpretation of the relative contributions of different regions to vertically integrated BC in Fig. 13. The meaning of relative seasonal variations in Fig. 13 is not clear without this information.

P. 12, L. 10: Please describe what is meant by a "sensitivity analysis". Were simulations repeated with modified/masked emissions using the same meteorology? Did the approach account for non-linear interactions of aerosols and trace gases from different regions? Did you verify that BC concentrations from individual model experiments with modified emissions add up to the concentrations obtained by running the model with all emissions included?

P. 13, L. 16: Fig. 12 seems to show key results for this study. However, the figure is confusing and needs to be cleaned up and explained. Several sub-panels are included but not explained. Would it be possible to include a table with annual mean results in order to summarize some of the results in this figure more clearly? Also, it would be very useful to compare results to multi-model results from AMAP (2015) for 2010. According to AMAP (2015), domestic emissions from East and South Asia are the largest source of annual mean BC in the Arctic. Emissions from Russian fires, Russian oil and gas flaring are also important according to AMAP (2015). Contributions of emissions from Nordic countries and the rest of Europe are much less important than contributions from Russian emissions. However, results in Fig. 12 show similar relative contributions for European and Russian emissions. Is it possible that a lack of Russian oil and gas flaring emissions in WRF simulations might account for this difference? Furthermore, the WRF model domain does not seem to include emissions of BC from fires in Africa and South America. According to AMAP (2015), contributions of BC emissions from the rest of the world (i.e. mainly Africa and South America) to Arctic BC burdens are similar to contributions of BC emissions from North America and Europe. Overall, it seems that results from AMAP (2015) are quite different from results simulated by WRF.

P. 13, L. 28: Vague. Overall, the context of the study is not clear (air quality vs climate processes).

P. 14, L. 22: The title of the section is misleading since no transport pathways are actually analysed in this section. It is not clear how the discussion of concentration cross sections helps to explain the transport of aerosols to the Arctic without information about the simulated circulation. Typically, advection leads to import of BC mass to the Arctic in some regions and export in other regions (e.g. see Iversen, 1996).

P. 15, Section 4: A critical evaluation of results and comparisons with results from other studies is missing. Contributions of emissions from different geographic regions and emission sectors have been studied before (see references in introduction and review comments above). It is not obvious whether the current study adds any significant new results?

P. 29, Fig. 1: Please indicate the base year of the emissions. It seems that emissions due to oil and gas flaring are not accounted for? These emissions are an important source of BC in the Arctic (AMAP, 2015).

P. 31, Figs. 3 and 4: These figures are confusing and need to be modified or replaced. First, very large regions with considerable meteorological and chemical variations are covered by each individual flight track so that mean results for individual flights are not very meaningful and difficult to compare with each other. Second, what is the purpose of using a time axis? Can results for individual days be used to understand the temporal evolution of plumes of polluted air? Finally, the large amount of information

[Figure]

Interactive
comment

in the figure is overwhelming. A much better way to present the data would be to produce plots similar to Fig. SM2, with meteorological variables plotted along flight tracks. Furthermore, it is not obvious that a comparison of meteorological results is necessary for the validation of simulated concentrations? Perhaps these figures can be moved to the supplementary document?

P. 42, Fig. 14: Why are results for dust shown in this figure? Dust has not been addressed before in the paper. In particular, it is not clear how well the model simulates dust. Furthermore, the labels in the figure are much too small and so it is not clear what longitude range is shown. Locations of Eurasia and Siberia are not clear. What are the units of the concentrations?

Technical corrections:

There are several grammar mistakes. A few examples are given in the following. It will be necessary to check the text for additional grammar mistakes.

P. 5: L. 9: "been" missing. L. 10: remove "the" L. 11: "an" missing. L. 16: "detail" L. 15: "The model used..." (past tense) vs "The model includes..." L. 15: missing "a"

P. 7, L. 27: Redundant information. Tables 1 and Fig. SM2 were already mentioned in the previous section.

P. 8, L. 19: A reference is missing for the MAC value used in the current study.

P. 14, L. 25: "Concentration" instead of "magnitude".

P. 36, Fig. 8: Values on the map are mentioned but are not actually visible in the plots?

P. 38, Fig. 10: What is OPM2.5?

P. 40, Fig. 12: Abbreviations BBSI, BBEU etc. in the figure need to be explained in the caption. It is not clear what these represent.

References:
AMAP: AMAP assessment 2015: Black carbon and ozone as Arctic climate forcers, Oslo, Norway., 2015.

Bond, T. C., et al. (2013), Bounding the role of black carbon in the climate system: A scientific assessment, J. Geophys. Res. Atmos., 118, 5380–5552, doi:10.1002/jgrd.50171.

Browse, J., Carslaw, K. S., Arnold, S. R., Pringle, K., and Boucher, O.: The scavenging processes controlling the seasonal cycle in Arctic sulphate and black carbon aerosol, Atmos. Chem. Phys., 12, 6775-6798, https://doi.org/10.5194/acp-12-6775-2012, 2012.

Iversen, T., 1984. On the atmospheric transport of pollution to the Arctic. Geophysical Research Letters, 11:457-460.

Mahmood, R., K. von Salzen, M. Flanner, M. Sand, J. Langner, H. Wang, L. Huang, 2016, Seasonality of Global and Arctic Black Carbon Processes in the AMAP Models, J. Geophys. Res., doi:10.1002/2016JD024849.

Sharma, S., M. Ishizawa, D. Chan, D. Lavoue, E. Andrews, K. Eleftheriadis, and S. Maksyutov (2013), 16-year simulation of Arctic black carbon: Transport, source contribution, and sensitivity analysis on deposition, J. Geophys. Res. Atmos., 118, 943–964, doi:10.1029/2012JD017774.

Stohl, A., Aamaas, B., Amann, M., Baker, L. H., Bellouin, N., Berntsen, T. K., Boucher, O., Cherian, R., Collins, W., Daskalakis, N., Dusinska, M., Eckhardt, S., Fuglestvedt, J. S., Harju, M., Heyes, C., Hodnebrog, O., Hao, J., Im, U., Kanakidou, M., Klimont, Z., Kupiainen, K., Law, K. S., Lund, M. T., Maas, R., MacIntosh, C. R., Myhre, G., Myriokefalitakis, S., Olivie, D., Quaas, J., Quennehen, B., Raut, J.-C., Rumbold, S. T., Samset, B. H., Schulz, M., Seland, O., Shine, K. P., Skeie, R. B., Wang, S., Yttri, K. E., and Zhu, T.: Evaluating the climate and air quality impacts of short-lived pollutants, Atmos. Chem. Phys., 15, 10529-10566, https://doi.org/10.5194/acp-15-10529-2015,

2015.

---

## Author Comment (AC1) · 16 Jun 2018

**Response to Reviewers**

**Source Sector and Region Contributions to Black Carbon and PM$_{2.5}$ in the Arctic**

Negin Sobhani[1,2], Sarika Kulkarni[2,3], and Gregory R. Carmichael[2]

[1] National Center for Atmospheric Research, Boulder (NCAR), Colorado, USA
[2] Center for Global and Regional Environmental Research (CGRER), University of Iowa, Iowa City, Iowa, USA
[3] California Air Resources Board (CARB), Sacramento, California, USA

*Correspondence to*: Negin Sobhani (negins@ucar.edu)

June 13, 2018

Dear Dr. Kathy Law,

We thank you and the reviewers for your time and the constructive comments and suggestions for improving this manuscript. We have carefully reviewed the manuscript and addressed all the comments. Below is the point by point responses to the reviewers' comments with the corresponding changes and refinements made in the revised paper. The reviewers' comments are shown in gray, the responses are shown in black plain text. We have made our best effort to accommodate all suggestions in details. The modified or added text in the manuscript is shown in *italics* with quotation marks. We have also included a highlighted version of the manuscript in PDF format to communicate changes and modifications effectively and easily.

Thank you for your consideration of this submission. We hope our responses adequately address the reviewers' comments and suggestions. We would highly appreciate your time and comments and look forward to your decision.

Best Regards,
Negin Sobhani, on behalf of all co-authors

**Reviewer I**

This study quantifies the contributions of different emission regions and sectors to Arctic Black Carbon (BC) and PM2.5 concentrations using a chemistry transport model. First, the authors evaluate their model to the ARCTAS flight campaign and two stations in the Arctic. Then, by using a sensitivity analysis they identify the main sectors contributing to Arctic PM2.5 (the power, industrial and biomass burning sector), and the largest contributors to BC surface concentrations (the residual and transport sector). Further, they look at the seasonal cycle and emissions from Europe and China in particular. The study is well-written and concise and the figures are clear and easy to understand. I really appreciate the seasonal focus in this study. I recommend this manuscript for publication after some clarifications, more details given below.

**Response:**  We really appreciate your constructive comments and suggestions for improving this paper. We have provided itemized responses to your comments and suggestions.

**1. Methods**

- Emissions. Can you please add a paragraph describing the emission dataset in more detail instead of just referring to Kulkarni et al. 2015? E.g. can you explain the different sectors? I like the plots of the data showing the geographical distribution (Fig 1), but could you also say something about the seasonal cycle? Do all sectors have a seasonal cycle? How is this calculated? E.g. is there a correlation with outdoor temperature in the residual sector? Is the seasonal cycle the same every year except for biomass burning? Could you say something about the uncertainties in the emissions?

  **Response:** Thanks so much for these comments and questions on emissions.  The following two paragraphs including more details about the emissions used in this study, has been added to section 2.1.1.

[revised manuscript text omitted]

- STEM model coupled to WRF. Could you also add a paragraph describing this model? Are there any known biases?

**Response:** Thank you for this comment. The following paragraphs are now included in section 2.1.3. for additional information on the STEM model.

*"This model is investigating the convective-diffusion equation below with Eulerian approach to calculate the concentration of chemical species i ($c_i$).*

$$\frac{\partial c_i}{\partial t} + \nabla(v c_i) = \nabla.K.\nabla c_i + R_i + S_i + G_i$$

*In the above equation,  $c_i$ is the gas phase concentration of  compound i, $v$ is the wind velocity vector, $K$ is for  eddy diffusity tensor,  $R_i$ is the total reactions of species i , $S_i$ denotes the sources for species i and $G_i$ is the mass transfer between gas and liquid (Kulkarni, 2009). The dry deposition of particles was calculated based on the resistance in series parameterization developed by Wesely and Hicks, 2000 and the values vary with meteorological conditions and land cover (Adhikary et al., 2007; Kulkarni et al., 2015; Sobhani, 2017; Uno et al., 2004). Wet deposition was modeled as a function of loss rate based on the meteorological fields (precipitation rates) from the WRF model as described in Uno et al., 2003 and Adhikary et al., 2007. Aging has been considered for both BC and OC particles using 7.1e-6 s as the aging rate (Adhikary et al., 2007; Cooke and Wilson, 1996). In this study, we used STEM model for simulating BC, OC, sulfate ($SO_4$), $SO_2$, $PM_{2.5}$, $PM_{10}$ , and other primary emitted $PM_{2.5}$ and $PM_{10}$ ."*

*"Regional models such as STEM require initial and boundary conditions from a larger-scale model to achieve reasonable predictions (Abdi-Oskouei et al., 2018; Tang et al., 2007). The STEM model used fixed boundary conditions for these annual simulations. The boundary conditions varied spatially and vertically based on observations from previous aircraft field experiments and discussed in detail in Tang et al., 2004. Further details describing this modeling system can be found in Kulkarni et al., 2015 and D'Allura et al., 2011."*

STEM has been successfully used in several field campaigns before including TRACE-P/ACE-Asia, ICARTT, INTEX-B, ARCTAS, PACDEX and others. There are no known biases in STEM model that the authors are aware of.

- How is the model setup regarding the sensitivity analysis? How did you perform the experiments? Simulation period? Please add a paragraph describing this as well.
  **Response:**

The following section (2.1.4.) has been added to the paper to describe the sensitivity simulations conducted for this study:

*"2.1.4. Sensitivity Simulations*

*For making effective emission mitigations policies, it is essential to assess the impacts of source sectors and source regions on the Arctic pollution. The base simulations and sensitivity analysis with perturbed emissions were performed to assess the impacts of various emission sectors and regions on the concentrations of PM and its components in the Arctic. The sector contributions were calculated using a series of model runs by eliminating the emissions of a particular sector each time. The base simulation included emissions from all sectors and used meteorology from the WRF model for the study period. The contributions of each sector to the PM concentrations were calculated as the difference between the base case and a simulation including all emissions but zeroing out the specific sector. Additional simulations were performed to calculate the source contribution from specific regions to PM concentrations over the Arctic. Using a similar method, sensitivity simulations were also performed to estimate the contributions of economic sectors from each of the geographic source regions to the Arctic surface and column concentrations. "In all case we used a zero out perturbation. These large emissions changes can lead to errors in secondary pollutants if the chemistry is non-linear. As BC, dust, and primary PM are primary species the results are not sensitive to non-linear effects. Sulfate is a secondary pollutant but its chemistry (in cloud and gas phase) is treated as a linear process in this model experiment."*

The simulation period for all simulations were April 2008- August 2009.

**2. **Figures:**

- In general, I think the captions could contain more information about the data shown (and what the boxes, lines etc. represent).
  **Response:** Thank you for your suggestions to improve the quality of the figures. We have revised all figures and their captions carefully. We have added more information about the plots representations and acronyms used in figures in the corresponding figure captions.
- FIG3: I suggest moving figure 3 to the Supplementary.
  **Response:** Thank you for this suggestion. We have moved Figure 3 to the supplementary materials (Fig SM3).

- FIG4: Could you add more text in the manuscript on Figure 4? What does the boxes represent? Is it each flight averaged over the column? Maybe you could replace this figure with SM3, which is easier to read I think. Or is your point here to show the flight by flight variation?

  **Response:** Thank you for this comment. We have added this to the caption of this figure and other boxplot figures in this paper:

  *"In box and whisker, the middle line denotes the median value, while the edges of the box represent 25$^{th}$ and 75$^{th}$ percentile values respectively. The whiskers denote the maximum and minimum values."*

  The model data were extracted along the flight paths for this comparison. The blue boxes and whiskers show the distributions for simulated values along the flight paths. In Fig SM3 (SM4 in the current version) both modeled and observed values are binned by flight altitudes every 1 km and plotted as vertical profiles. Since the vertical profile of the flights are already mentioned in Fig 5 for all flights, we preferred to keep Fig SM3 in the supplementary materials.

- FIG5: Again, can you explain what the boxes and whiskers represent? The model, as you say, is biased high at higher altitudes. This is common for many models. Could you maybe speculate a little on the reasons for why your model overestimate BC at high altitude, as you do for SO4?

  **Response:** Thank you for this excellent comment. We have added the following statements to the manuscript regarding the reasons of BC overestimation at higher altitudes.

  *"Pollutant transport across the Pacific happens in discrete plumes during Springtime (Adhikary et al., 2010). CTMs tend to disperse these plumes in vertical layers of the atmosphere too much. This spreading typically results in decreases in modeled peak values (Adhikary et al., 2009; Kulkarni, 2009). The underestimation of BC at the surface may also be attributed to an underestimation of BC emissions especially at higher latitudes e.g., gas flaring (Huang et al., 2014, 2015; Stohl et al., 2013) and shipping emissions (Marelle et al., 2016). The over-prediction of BC at higher altitudes might be due in part to underestimations of BC removal by frozen clouds and precipitations (Koch et al., 2009)"*

  Similar to the above comment, more information on box and whisker plots are added to this plot.

- FIG6: Impressive seasonal cycle in the model. Not many models have such a distinct seasonal cycle. To be clear: in the data at Barrow used here the BB contributions were removed, right?

  **Response:** Thanks for this comment. To reduce the local contamination, the observational data from Barrow site are routinely screened. Stohl et al, 2006 found that the biomass burning plumes are unintentionally removed at Barrow site. Based on this screening only values are considered valid when the wind direction is between 0-130 (Bodhaine, 1995; Hirdman et al., 2010) . Stohl et al. 2013 study mentions that by reducing the data screening, the EBC values were increase by a factor 2-3 in summer.

- FIG7: Could you make the dots representing the IMPROVE observations larger? It is a bit hard to spot them. In the text you say that this is 'annual mean', but the simulation period is April 2008 to July 2009. How did you average the data? Over the whole period? Is 0.16 ug/m3 averaged over the entire US in the model? If so, is that number very different by just averaging the grid boxes containing stations? Are the IMPROVE monthly or daily data? How did you compare missing data etc?

  **Response:** We have updated and improved Figure 7 as you suggested. We changed the colormap for better clarity and made the stations 50% bigger. We have also increased the thickness of black line circling each station for better clarity and visibility.

  The annual mean is for April 2008-Mar 2009. The model shows all simulated data average for April 2008-Mar 2009. The observation values show the average available data for each station. IMPROVE data has daily frequency. Annual means are calculated based on averaging daily observation from April 01, 2008 to March 30, 2009. Since the frequency of missing data is different at each station, we used all available data points. The modeled annual means are based on averages of all data from April 2008- Mar 2009.

- FIG8: The caption says 'Dust' for the 'SO4' plot. Again, you say in the text that this is annual average, but in the caption that this is an average over the simulation period?

  **Response:** Thank you so much for this comment. We fixed the error and exchanged dust with sulfate (SO4) in figure 8's caption. This figure shows the annual average for one year starting from April 01, 2008 to March 31, 2009. This description is also added to the caption of this figure.

- FIG10: OPM2.5? This is the first time we see OC, and it is not mentioned in the text.

**Response:** Thank you for this comment. OPM2.5 is the acronym for other PM2.5 and refers to other primary emitted non-carbonaceous particles with aerodynamic diameters less than 2.5 µm. We have also added this information to Figure 10's caption.

"*OPM$_{2.5}$ is the acronym for other PM$_{2.5}$ and refers to other primary emitted non-carbonaceous particles with aerodynamic diameters less than 2.5 µm such as fly ash, road dust, and cement which were simulated as a single mass component in the model.*"

- FIG14: The row with the Dust plot seems a bit out-of-place here. At least change the 'Base' to BC in the first row and have the same color scale as dust? Is this fine-mode dust as in FIG10.

**Response:** Thank you for this suggestion. We have changed Base to BC in the model's title. We have improved the quality of Fig 14 and make x axis labels (latitude and longitudes) font larger. We have removed the last row (dust) and moved it to SM (Figure SM12). The dust row (now removed) of this figure is for comparing transport of dust vs. BC. Dust major emission sources are lower latitudes and Central Asia, compared to BC emission sources. Please note that dust denotes sum of both fine and coarse dust concentrations.

**3. Conclusions:**

- Last paragraph: Could you be a bit more specific here? I miss a discussion on the uncertainties in the emission inventories, observations and the data you have used, but that can be written elsewhere e.g. in the Methods section. You highlight high-resolution modeling; why would high resolution modeling studies be the most necessary to reduce the uncertainties in the future? Do your conclusions fit well with what other studies have shown? What are the implications of your study?

**Response:** Thank you for this comment. We have added discussions about uncertainties for emissions inventories (both introduction and method) and for observations (in method section) used in this study. We have made several modifications to the conclusions section to address this comment. We tried to incorporate the related literature as much as possible in the results section. However, a quantitative inter-comparing between our results and previous results is challenging due to multiple reasons including difference in region definitions, emission inventories, simulation periods.

**4. Minor Comments:**

- Page 2, L8: With 'albedo reduction' do you mean the general warmer temperatures → sea-ice is melting → more open water, or the albedo reduction caused by BC deposition only? If the latter, as it is written now it seems like the largest feedback in the Arctic causing the doubling of temperature increase is caused by BC.

  **Response:** Thanks for this comment. This sentence is slightly modified for clarity.

  Also, exactly two lines after this sentence, it is mentioned than ~25% of the past century's warming over the Arctic is caused by BC.

- Page 4: L1: You say here that you studied organic carbon? As far as I can see, there is no mentioning of OC except that it is included in the bottom panel of FIG10. And what about dust?
  **Response:** Thanks for this comment. OC is studied only as a component of $PM_{2.5}$ in this paper. The focus and evaluations were mostly concentrated on BC and sulfate.
- Page 9, L21-23: Where is this shown? Can you point us to a figure?
  **Response:** This is shown in figure 10 and figure 12. Both figures contain information on the time series contributions of residential sector to Arctic BC.
- Page 9, L24: What is the average in the Arctic region? You compare your range to Sharma et al 2013; can you add a sentence about what this number (0.06 ug/m3) represents (e.g. all surface stations?)?
- **Response:** Thanks for your comment. We have added the following sentence including the Arctic are average (> 60°N )to this paragraph for clarification:
  *"The simulated annual BC average for the Arctic area (latitudes > 60°N) is on average ~ 0.65µg/m3. "*
  This value represents the annual average for all grid points with latitudes > 60°N.

- Page 9, L28: Can you add chapter reference for IPCC instead of just the whole report?
  **Response:** This has been implemented.
- Page 10, L10: Can you add why (~ the natural sources are larger in size)?
  **Response:** Significant components of PM10 are coarse dust and sea salt (both emitted via natural sources) while for PM2.5, the major components sulfate, BC, and OC (anthropogenic sources). The significance of sea salt to PM 10 over oceans has been discussed in the same paragraph.
- Page 10, L18: 'global warming' or you only mean (local) warming over the Arctic?

**Response:** Thank you for this comment. For clarifications we changed this sentence to the following statement:

*"Due to the significant contribution of BC in warming over the Arctic and its amplification mechanisms, it is important to understand the influence of specific source regions and source sectors on the Arctic BC concentration."*

- Page 11, L1: Could you add numbers?

**Response:** Thank you. The values are added to this line.

- Page 12, L25. Move or remove the parenthesis

**Response:** Thank you. This has been implemented.

- Page 15: L4: I guess you refer to 14-l (not 14-i)

**Response:** Thank you. This has been corrected.

**Reviewer II**

**General comments:**

The study adds to a growing list of studies on impacts of regional emissions of black carbon on Arctic aerosol concentrations. Given large yet uncertain impacts of aerosols on Arctic climate, there is considerable need for research in this area. The authors of the study analysed at a wide range of model results in great detail. However, there are substantial issues with the design of the study and presentation of the results.

**Response:** Thank so much for your comments and suggestions. We have provided itemized responses to your comments below.

- First, important sources of black carbon were not accounted for. Consequently, BC concentrations in the Arctic are underestimated and conclusions about the relative importance of different source regions are biased. More specifically, emissions of black carbon from central and south America, central and south Africa, and Australia are not accounted for in the simulations. According to data sets that were used by AMAP for an assessment of the impacts of black carbon on the Arctic (AMAP, 2015; Stohl et al., 2015), these regions contributed about 41% to total (anthropogenic and natural) global emissions of black carbon in 2010 (AMAP region "ROW"). The simulated contribution of these emissions to total black carbon mass in the Arctic atmosphere is 10-20%, depending on the model. Furthermore, it appears that emissions associated with oil and gas flaring were not accounted for. According to AMAP data, about 65 GgC of black carbon were emitted by oil and gas flaring industrial activities in 2010, especially from sources at high latitudes. Efficient transport to the Arctic implies that the contribution of these emissions to total black carbon mass in the Arctic atmosphere is relatively large, i.e. comparable to the impacts of emissions of black carbon from all North American sources.

  **Response**: Thanks for this comment. First, our modeling domain were extended over northern Africa, Middle East, and south Asia to include the emissions from population-dense regions, while in AMAP 2015, ROW (Rest of the world regions) is sum of central Asia, Middle East, northern Africa, southern Asia, and southern hemisphere emissions (AMAP 2015, Figure 5-1). The STEM model used fixed boundary conditions for these annual simulations. The boundary conditions varied spatially and vertically based on observations from previous aircraft field experiments and discussed in detail in (Tang et al., 2004). However, no emission perturbation simulations were done on the BC coming from the boundaries.
  Furthermore, Shekar and Olivier, 2007 shows that emission from Australia, South America, and Africa each contribute to ~1% of sum of Arctic and Antarctic BC surface deposition. Hence, we can assume that their contributions to Arctic BC is <1%. Similarly,

Zhang et al., 2015 shows that contributions of emissions from Australia and South America is 0% and <0.1% to sum of Canada, former Soviet Union and Europe BC burden.

We have added the following statements to the manuscript for clarifications:

*"Regional models such as STEM require initial and boundary conditions from a larger-scale model to achieve reasonable predictions (Abdi-Oskouei et al., 2018; Tang et al., 2007). The STEM model used fixed boundary conditions for these annual simulations. The boundary conditions varied spatially and vertically based on observations from previous aircraft field experiments and discussed in detail in Tang et al., 2004. Further details describing this modeling system can be found in Kulkarni et al., 2015 and D'Allura et al., 2011."*

We agree that under-estimation of flaring emissions could lead to the under-estimations of surface and column concentrations over the Arctic. However, a recent study (Winiger et al., 2017) using Bayesian approach, FLEXPART , and 2-yr continuous observations identified the errors in space allocation and suggested -84% reduction in this emission, which translates to (6.25x) overestimation in the previous emission inventory. We have added the followings to the manuscript (introductions and results sections) for clarifying this important point:

*"For example, while previous studies estimated that oil and natural gas flaring is an important sector contributor to the Arctic (AMAP, 2015; Eckhardt et al., 2015; Huang et al., 2014, 2015; Sand et al., 2015; Stohl et al., 2013; Xu et al., 2017), a recent paper (Winiger et al., 2017) showed that emissions from oil and gas flaring contribute to only ~6% of Arctic BC concentration indicating a 6.25× overestimation of flaring emissions in the previous studies."*

*"Previously, Stohl et al., 2013 study suggested that emission from oil and natural gas flaring in Russia is an important but overlooked source of Arctic BC contributing to % 66 of total anthropogenic emissions within the Arctic (latitudes > 66 °N). Similarly,* Huang et al., 2015 estimated that gas flaring emissions accounts for 36.2% of total anthropogenic BC emissions from Russia. *Using similar emission inventory, AMAP, 2015, Eckhardt et al., 2015, Huang et al., 2014, Huang et al., 2015, Sand et al., 2015, Stohl et al., 2013 ,and Xu et al., 2017 concluded that flaring is a significant contributor to Arctic BC. However, a recent study* (Winiger et al., 2017) *using Bayesian approach, FLEXPART, and 2 year continuous observations identified the errors in space allocation and suggested -84% reduction of flaring emissions, which translates to (6.25x) overestimation of flaring emissions in the previous emission inventory. Winiger et al., 2017 study shows that contribution of gas flaring is relatively small (6%) compared to*

*residential (35%) and transport (38%) sectors, which is similar to our results showing residential and transportation are contributing ~38% and ~30% to the Arctic BC."*

2) Second, the analysis of model results in the study seems narrowly focused on results for the Arctic, which is problematic with regard to an improved understanding of the impacts of aerosols on Arctic climate. For instance, aerosol radiative effects at mid latitudes have a strong impact on Arctic climate by influencing the transport of heat to the Arctic. There are also impacts of black carbon on snow albedos, which are not considered either.

**Response:** It is important to note that the focus of this study is not studying the impacts of BC on radiative forcing or climate. The goal of this study is to quantify the contributions of emission sources from different economic sectors and geographical region to the Arctic pollution. For developing effective emission mitigation policies, it is necessary to quantify sources and economic sectors and identify the transport pathways of pollutants to the Arctic. We have added the following paragraph to the manuscript for better clarifying the scope of this study:

*"It is crucial to identify the sources of Arctic pollution in order to devise effective control strategies for mitigating the Arctic air quality, climate, and radiation imbalances. The primary goal of this study is to quantify the relative contributions of different source sectors and source regions on the arctic aerosol concentration (surface and column abundances) and its impact on Arctic air quality through a series of model sensitivity simulations. Although the aerosol vertical profiles and column abundances are discussed, addressing the aerosol radiative and climate impacts is beyond the scope of this work."*

3) Finally, it is not obvious why results for PM2.5, PM10, sulphate, and dust concentrations are analysed. Comparisons with observations are missing and analysis of model results for these is less complete than the analysis for black carbon. Overall, the relevance of model results is not obvious with regard to impacts of aerosols on climate or air quality. There is no discussion of climate implications or comparison with results from other studies.

**Response:** As mentioned in the above comment, the goal of this study is not calculating the radiative forcing or climatic impacts caused by aerosols, but to understand which source sectors and source regions are contributing the most to the Arctic surface and column PM concentrations. The results and discussions of this study are mostly focused on BC and sulfate (modifications made accordingly). Additional evaluations for sulfate at two surface sites (Alert and Barrow) are added to the discussion. Correspondingly the information about the sulfate measurements is added to the method section and results

section. Figure 6 and the following paragraphs has been added to the manuscript to address this comment.

*"Sulfate measurements at Barrow and Alert are taken using ion chromatographic analysis (Hirdman et al., 2010; Quinn et al., 1998; Sirois and Barrie, 1999). A basic high volatile sampler from Sierra Instruments is used for collecting aerosol samples at both the monitoring sites. The measured sulfate concentrations at both Alert and Barrow sites were corrected by subtracting sea salt component using aerosol sodium (Na+) and chlorine (Cl-), which is mostly from marine sources (AMAP 2015; Barrie and Hoff, 1985; Hirdman et al., 2010; Quinn et al., 1998, 2000). Therefore, the reported non-sea salt (nss) sulfate can be directly compared with the modeled values. It should also be noted that the sample durations for Alert and Barrow sites varied 1-5 days for Barrow and 3-9 days for Alert (Hirdman et al., 2010; Quinn et al., 1998; Sirois and Barrie, 1999)."*

*"**Error! Reference source not found.** shows monthly boxplots comparing simulated sulfate with observed values at Alert and Barrow. Both stations show strong seasonal variation with the minimum occurring during summer and early fall similar to BC. As discussed above for BC, this is due to the northward retreat of the Arctic front and efficient wet scavenging during summer. The model accurately captured the seasonality of observed sulfate at both sites. The summer-time minima of sulfate reflects the less effective transport and high scavenging during summer. At Barrow, the model over-predicted the observed values throughout the year. However, during spring and winter, the simulated sulfate values are much closer to the observation. It should be noted that the observations from Barrow site has large data gaps and missing data possibly due to equipment malfunction. To avoid local contamination, the sector source controlled sampling method removes data suspected to be contaminated by the town of Barrow (Bodhaine, 1989, 1995; Fisher et al., 2011; Hirdman et al., 2010a). The significant data gaps might introduce biases in the monthly calculations. There were also no sulfate measurements for July, August, and December 2008. The model over-predicted sulfate at the Alert site, except for winter-time. During winter, the model accurately predicted the range of simulated sulfate at the Alert site. The over-prediction during summer might be due to the less effective scavenging processes and higher magnitude of transport in the model. The results are similar to the (Hirdman et al., 2010b) study, which used nss sulfate monthly averages dung the years 2000-2006. Observations and model show that Barrow shows much higher concentrations of sulfate throughout the year when compared to the Alert site."*

It should be noted that We included dust results since dust can be an important component of Arctic PM with high impact on PM seasonal cycle. For dust, the model evaluation over the Arctic is limited due to a scarcity of observations. Satellite data

(MODIS) can only identify high dust events over the Arctic and provide no meaningful quantitively data (Groot Zwaaftink et al., 2017). Using a similar modeling setup with the same dust emissions, we evaluated AOD extensively on hemispheric level with MODIS satellite observation, AERONET, EANET, and EMEP sites in Kulkarni et al., 2015 Supplemental Material (https://www.atmos-chem-phys.net/15/1683/2015/acp-15-1683-2015-supplement.pdf).

**Specific comments:**

- P. 2, L. 22: Another great reference is Bond et al. (2013).

  **Response:** Thanks for this comment. Bond et al. 2013 is added as the reference for this sentence.

- P. 3, L. 10: Sharma et al. (2013) and references given in that paper seem relevant here, too.

  **Response:** Sharma et al. 2013-a, Wang et al. 2014, Ikeda et al. 2018 are added as additional references for the next sentence discussing the studies showing significant contribution of Asian emissions to the Arctic.

- P. 3, L. 27: Uncertainties and biases in parameterizations of wet deposition and convective processes should be more emphasized here since they mainly explain differences in simulated aerosol concentrations in the Arctic. See Browse et al. (2012) and Mahmood et al. (2016) and references in these papers.

  **Response:** Thank you for your comment. We have added the following statements discussing uncertainties and biases in further details.

  *"The modeling inter-comparison study by Eckhardt et al., 2015 showed that current models (including both atmospheric chemistry transport and climate models) were unable to reproduce the observed BC seasonality at the surface. There are also high discrepancies among different models in capturing BC concentrations over the Arctic (Eckhardt et al., 2015; Shindell et al., 2008), which is caused by various factors including emissions, meteorology, and transport patterns. The uncertainties associated with emissions is a key component of this inter-model variability and differences between simulations and observations. According to Ramanathan and Carmichael, 2008, regional emissions can have a factor of 2 to 5 uncertainty. For example, while previous studies estimated that oil and natural gas flaring is an important sector contributor to the Arctic (AMAP, 2015; Eckhardt et al., 2015; Huang et al., 2014, 2015; Sand et al., 2015; Stohl et al., 2013; Xu et al., 2017), a recent paper (Winiger et al., 2017) showed that emissions from oil and gas flaring*

*contribute to only ~6% of Arctic BC concentration indicating a 6.25× overestimation of flaring emissions in the previous studies.*

*Other factors that also contribute to the model-observation offset in the Arctic region are the uncertainties and errors in meteorology and transport mechanism (Jiao et al., 2014). Finally, the representation of the particle processes in the atmosphere is another major source of uncertainty in the inter-model variability. Errors and uncertainties in dry and wet removal processes (including in-cloud and below-cloud mechanisms) at high altitudes is a major source of uncertainty. Mahmood et al., 2016 study indicates that scavenging of BC in convective clouds outside the Arctic, substantially influences BC vertical distributions and overall wet deposition efficiency within the Arctic; hence, is one of the major cause of discrepancies in Arctic BC burdens among different models used in Eckhardt et al., 2015. Marelle et al., 2017 indicates that both surface and tropospheric BC in the Arctic are highly sensitive to the representation of cumulus cloud processes impacting aerosols."*

It is worth noting that several other factors including differences in meteorological conditions used by models and biases in the emissions inventories are the main reason of discrepancies in different model simulations to the biases and uncertainties of simulated aerosols as discussed above.

- P. 3, L. 28: The statement that regional chemical transport models capture BC concentrations better is highly questionable. According to Eckhardt et al. (2015), there is no single class of models that outperforms other models. The statement here is based on 2 individual models.

  **Response:** Thank you for this comment. That is correct. We have removed this sentence from the manuscript and added the following sentences:

  *"Eckhardt et al. 2015 inter-comparison modeling study showed that current models (including both atmospheric chemistry transport and climate models) failed in reproducing the observed BC seasonality at the surface level. There are also high discrepancies among different models in capturing BC concentrations over the Arctic (Eckhardt et al., 2015; Shindell et al., 2008)."*

- P. 3: The discussion in this section is focussed on simulations of surface concentrations. For climate and radiative forcings, aerosol vertical distributions and deposition of BC on snow are more important than surface concentrations. Please clarify the focus of the study on air quality aspects of Arctic aerosols or include further information regarding aerosol vertical profiles. This is particularly important since comparisons for aerosol vertical profiles are included in the paper (Section 3.1.2). The title and abstract are somewhat ambiguous in that regard.

**Response:** The focus of this study is identifying and assessing the impact of sectors and regions on the Arctic PM surface and column concentration. The scope of this study has been discussed in details in response to comment 2 of reviewer 2. Also, the following statements are added to the manuscript for clarifying the scope of this study.

*"It is crucial to identify the sources of Arctic pollution in order to devise effective control strategies for mitigating the Arctic air quality, climate, and radiation imbalances. The primary goal of this study is to quantify the relative contributions of different source sectors and source regions on the arctic aerosol concentration (surface and column abundances) and its impact on Arctic air quality through a series of model sensitivity simulations. Although the aerosol vertical profiles and column abundances are discussed, addressing the aerosol radiative and climate impacts is beyond the scope of this work."*

- P. 4, Section 2.1.2: Please indicate whether annual mean emissions are used. Are temporal variations in fire emissions are accounted for?

**Response:** Thank you for your comment. The emissions used in this study including both anthropogenic and biomass burning emissions have temporal resolution and the annual mean emissions are not used. The anthropogenic emissions (HTAP_v2.2) is based on Janssens-Maenhout et al., 2015 for 2008 and has monthly temporal resolution. The biomass burning emissions are from FINNv1(Wiedinmyer et al., 2011). FINN v1 emissions provides daily emissions with high resolution using satellite observations (MODIS) of fires and land cove changes. The followings with more information about the emissions are added to section 2.1.2.

*"The base emission setup used for this modeling study is similar to Kulkarni et al. 2015, except that anthropogenic emissions were updated to the Hemispheric Transport of Air Pollution phase 2 emissions inventory (HTAP_v2.2) for the year 2008 (Janssens-Maenhout et al., 2015). HTAP_v2.2 emission inventory contains comprehensive harmonized sector-specific 0.1° × 0.1° longitude-latitude emission grid maps for SO2, NOx, NMVOC, NH3, PM10, PM2.5, BC and OC with monthly and yearly temporal resolution for the years 2008 and 2010 (Janssens-Maenhout et al., 2015, data available at http://edgar.jrc.ec.europa.eu/htap_v2/index.php?SECURE=123). For this study, we utilized the monthly-varying emissions available for 2008 from HTAP v2.2 emission inventory. HTAP_v2.2 emission is based on a collection of different regional gridded emission inventories per sector and per region including that of the European Monitoring and Evaluation Programme (EMEP) and Netherlands Organisation for Applied Scientific Research (TNO) for Europe, the Environmental Protection Agency*

*(EPA) for USA, the EPA and Environment Canada (for Canada), and the Model Inter-comparison Study for Asia (MICS-Asia III) for China, India and other Asian countries (Janssens-Maenhout et al., 2015; Li et al., 2017; Lu et al., 2011). For the rest of the world (South America, Africa, Russia, and Oceania) the emission grid maps of the Emissions Database for Global Atmospheric Research (EDGARv4.3) bottom-up inventory was used in HTAP_v2.2 (Janssens-Maenhout et al., 2015). The sectors in HTAP_V2.2 emission dataset are based on IPCC 1996 categories definitions. In this study Energy (alternatively named as Power) sector is defined as total emissions from stationary and mobile energy activities for electricity generation, which includes fuel combustion as well as fugitive fuel emissions. The industrial sector includes emissions from industrial large-scale combustion emissions other than electricity productions (power sector) and emissions from industrial processes and solvent productions and applications. Emissions from the residential sector are from small-scale combustion including heating, cooling, illuminations, cooking and other auxiliary engines (such as lifting systems) to equip residential buildings, commercial buildings, agricultural facilities (including fisheries), waste-water treatment, and solid waste disposal and incineration."*

*"Emissions from the residential sector have strong seasonal (monthly) variations, that is negatively correlated to the temperature in most of the regions due to the use of heating systems (Janssens-Maenhout et al., 2015). In some developed countries, the residential sector emissions have a positive correlation with the temperature during the summer due to the increase in emissions from air conditioning devices (Janssens-Maenhout et al., 2015). Emissions from transport, industry, and energy sectors show modest seasonality in all regions (Janssens-Maenhout et al., 2015). There are high uncertainties in HTAP v2.2 PM2.5 and BC emissions emerging from different sources (especially transport and residential sectors). These uncertainties originate the uncertainties in officially announced annual inventories provided by countries, uncertainties due to process representation (the quality and representativeness of the controlled emission factors), uncertainties due to aggregations (grid maps used for allocating national totals for a source category will be different from the maps used at national levels). It is important to note that PM2.5, BC, and OC emissions from residential and transport sectors are qualitatively classified as highly uncertain in HTAP v2.2."*

Furthermore, for fire emission the following sentence is added:

*"FINN provide daily global emissions based on satellite (e.g., MODIS) observation for detecting active fires as thermal anomalies and land cover change (Wiedinmyer et al., 2011)."*

- P. 5, Section 2.1.3: This section lacks a lot of detail. What types of aerosols are consider and how they are numerically represented? Does the model simulate aerosol size distributions? What kind of aerosol processes are accounted for? Is aging of BC accounted for? How are interactions between aerosols and gas-phase chemistry represented?

**Response:** Thank you for this comment. More details and descriptions are added to section 2.1.3. :

*"This model is investigating the convective-diffusion equation below with Eulerian approach to calculate the concentration of chemical species I ($c_i$).*

$$\frac{\partial c_i}{\partial t} + \nabla(v c_i) = \nabla . K . \nabla c_i + R_i + S_i + G_i$$

*In the above equation, $c_i$ is the gas phase concentration of compound i, $v$ is the wind velocity vector, K is for eddy diffusity tensor, $R_i$ is the total reactions of species i , $S_i$ denotes the sources for species i and $G_i$ is the mass transfer between gas and liquid (Kulkarni, 2009). The dry deposition of particles was calculated based on the resistance in series parameterization developed by Wesely and Hicks, 2000 and the values vary with meteorological conditions and land cover (Adhikary et al., 2007; Kulkarni et al., 2015; Sobhani, 2017; Uno et al., 2004). Wet deposition was modeled as a function of loss rate based on the meteorological fields (precipitation rates) from the WRF model as described in Uno et al., 2003 and Adhikary et al., 2007. Aging has been considered for both BC and OC particles using 7.1E-6 s as the aging rate (Adhikary et al., 2007; Cooke and Wilson, 1996). In this study, we used STEM model for simulating BC, OC, sulfate, SO2, PM2.5, PM10, and other primary emitted PM2.5 and PM10."*

*"Regional models require initial and boundary conditions from a larger-scale model to achieve reasonable predictions (Abdi-Oskouei et al., 2018; Tang et al., 2007). The STEM model used fixed boundary conditions for these annual simulations. The boundary conditions varied spatially and vertically based on observations from previous aircraft field experiments and discussed in detail in Tang et al., 2004."*

- P. 7, Section 3.1.2: Comparisons for BC are included here but similar comparisons for sulfate are missing. The latter needs to be added (e.g. by comparing sulfate concentration in the Arctic with observations) because much of the subsequent discussion in the manuscript also addresses sulfate. It is not clear how well the model simulates non-BC aerosol species. Similar, it is not obvious that simulated results for

PM are realistic. Without validation it would be better to remove results for sulfate and PM from the paper altogether.

**Response:** Thank you. Comparisons of sulfate with observations along the ARCTAS flight paths have already been discussed in this section and in figures 4, 5, and SM3.

Additionally, we added the evaluations of sulfate at two Arctic surface concentrations. We made the boxplots for comparison of sulfate at Alert and Barrow. A new figure containing this has been added to the manuscript page. The following discussion has been added to the manuscript in method (section 2) and results (section 3.1.2.) regarding this comparison.

*"Sulfate measurements at Barrow and Alert are taken using ion chromatographic analysis (Hirdman et al., 2010; Quinn et al., 1998; Sirois and Barrie, 1999). A basic high volatile sampler from Sierra Instruments is used for collecting aerosol samples at both the monitoring sites. The measured sulfate concentrations at both Alert and Barrow sites were corrected by subtracting sea salt component using aerosol sodium (Na+) and chlorine (Cl-), which is mostly from marine sources (AMAP 2015; Barrie and Hoff, 1985; Hirdman et al., 2010; Quinn et al., 1998, 2000). Therefore, the reported non-sea salt (nss) sulfate can be directly compared with the modeled values. It should also be noted that the sample durations for Alert and Barrow sites varied 1-5 days for Barrow and 3-9 days for Alert (Hirdman et al., 2010; Quinn et al., 1998; Sirois and Barrie, 1999)."*

*"**Error! Reference source not found.** shows monthly boxplots comparing simulated sulfate with observed values at Alert and Barrow. Both stations show strong seasonal variation with the minimum occurring during summer and early fall similar to BC. As discussed above for BC, this is due to the northward retreat of the Arctic front and efficient wet scavenging during summer. The model accurately captured the seasonality of observed sulfate at both sites. The summer-time minima of sulfate reflects the less effective transport and high scavenging during summer. At Barrow, the model over-predicted the observed values throughout the year. However, during spring and winter, the simulated sulfate values are much closer to the observation. It should be noted that the observations from Barrow site has large data gaps and missing data possibly due to equipment malfunction. To avoid local contamination, the sector source controlled sampling method removes data suspected to be contaminated by the town of Barrow (Bodhaine, 1989, 1995; Fisher et al., 2011; Hirdman et al., 2010a). The significant data gaps might introduce biases in the monthly calculations. There were also no sulfate measurements for July, August, and December 2008. The model over-predicted sulfate at the Alert site, except for winter-time. During winter, the model accurately predicted the range of simulated sulfate at*

*the Alert site. The over-prediction during summer might be due to the less effective scavenging processes and higher magnitude of transport in the model. The results are similar to the (Hirdman et al., 2010b) study, which used nss sulfate monthly averages dung the years 2000-2006. Observations and model show that Barrow shows much higher concentrations of sulfate throughout the year when compared to the Alert site."*

- P. 11, L. 23: Figure 10 shows the seasonal cycle of aerosol concentrations at the surface. A similar figure needs to be added for the vertically integrated amount of BC in the Arctic. This information is necessary for the interpretation of the relative contributions of different regions to vertically integrated BC in Fig. 13. The meaning of relative seasonal variations in Fig. 13 is not clear without this information.

- **Response:** Thanks for your comment. Do you mean Fig. 12 instead of Fig. 13? Because Fig. 13 bar plots only show surface concentrations and are not related to column contributions. A figure similar to Fig. 10 for column BC (vertically integrated amount of BC ) is added to SM. The following statements are also added to this section regarding this.

  *"Figure SM8 shows the seasonal variation of contributions of different economic sectors to Arctic BC column concentration (vertically integrated amount of BC). The contribution of biomass burning to column concentration is very significant and much higher than the surface concentration in spring and especially during spring 2008. The heat and convection caused by the fires inject the biomass burning much higher in the atmosphere; hence the impact of biomass burning emission is accentuated in column concentrations. The biomass burning contribution to Arctic column BC in spring 2008 is almost double that of spring 2009, which shows the impacts of an unusually higher number of forest fires in 2008."*

- P. 12, L. 10: Please describe what is meant by a "sensitivity analysis". Were simulations repeated with modified/masked emissions using the same meteorology? Did the approach account for non-linear interactions of aerosols and trace gases from different regions? Did you verify that BC concentrations from individual model experiments with modified emissions add up to the concentrations obtained by running the model with all emissions included?

  **Response:** Thank you for this comment. The contributions of each sector of each region to the concentrations were calculated as the difference between the base case and a simulation including all emissions but zeroing out the specific sector of that region. The base case includes all emissions and uses a similar meteorology as the

other tests. The following section is added to the manuscript for more clarifications on sensitivity simulations:

*"2.1.4. Sensitivity Simulations*

*For making effective emission mitigations policies, it is essential to assess the impacts of source sectors and source regions on the Arctic pollution. The base simulations and sensitivity analysis with perturbed emissions were performed to assess the impacts of various emission sectors and regions on the concentrations of PM and its components in the Arctic. The sector contributions were calculated using a series of model runs by eliminating the emissions of a particular sector each time. The base simulation included emissions from all sectors and used meteorology from the WRF model for the study period. The contributions of each sector to the PM concentrations were calculated as the difference between the base case and a simulation including all emissions but zeroing out the specific sector. Additional simulations were performed to calculate the source contribution from specific regions to PM concentrations over the Arctic. Using a similar method, sensitivity simulations were also performed to estimate the contributions of economic sectors from each of the geographic source regions to the Arctic surface and column concentrations. "In all case we used a zero out perturbation. These large emissions changes can lead to errors in secondary pollutants if the chemistry is non-linear. As BC, dust, and primary PM are primary species the results are not sensitive to non-linear effects. Sulfate is a secondary pollutant but its chemistry (in cloud and gas phase) is treated as a linear process in this model experiment."*

- P. 13, L. 16: Fig. 12 seems to show key results for this study. However, the figure is confusing and needs to be cleaned up and explained. Several sub-panels are included but not explained. Would it be possible to include a table with annual mean results in order to summarize some of the results in this figure more clearly? Also, it would be very useful to compare results to multi-model results from AMAP (2015) for 2010. According to AMAP (2015), domestic emissions from East and South Asia are the largest source of annual mean BC in the Arctic. Emissions from Russian fires, Russian oil and gas flaring are also important according to AMAP (2015). Contributions of emissions from Nordic countries and the rest of Europe are much less important than contributions from Russian emissions. However, results in Fig. 12 show similar relative contributions for European and Russian emissions. Is it possible that a lack of Russian oil and gas flaring emissions in WRF simulations might account for this diffence? Furthermore, the WRF model domain does not seem to include emissions of BC from fires in Africa and South America. According to AMAP (2015), contributions of BC emissions from the rest of the world (i.e. mainly Africa and South America) to Arctic BC burdens are similar to contributions of BC

emissions from North America and Europe. Overall, it seems that results from AMAP (2015) are quite different from results simulated by WRF.

**Response:** Thanks for this comment. The caption for this figure is modified and more information on the acronyms are added to the caption on the acronyms. The figure is cleared up for better visibility.

Similar to AMAP 2015, the residential emissions from China is a significant source of annual mean BC column concentration. However, the regions are defined differently in AMAP 2015 compared to our study. AMAP 2015 study used ECLIPSE emission inventory which includes high emissions from flaring. In this emission inventory, flaring emission from Russia accounts for 93% of total anthropogenic BC emissions within the Arctic (Xu et al., 2017). Winiger et al., 2017 study reported errors in spatial allocations of BC sources over Russia and significant over-estimation of flaring emissions recommending -84% reduction of gas flaring emissions in ECLIPSE dataset. Generally, there are high uncertainties with the anthropogenic emissions from Russia. Huang et al. 2014 suggested the emissions from oil and gas industry and mining are highly under-estimated in EDGAR v4.3 (used in our study). The in-depth discussions on emissions from southern hemisphere are mentioned in the above comments (Comment 1 of Reviewer 2).

- P. 13, L. 28: Vague. Overall, the context of the study is not clear (air quality vs climate processes).
  **Response**: The context and scope of the study is discussed in the response for the comments above in details. Please refer to responses above.

- P. 14, L. 22: The title of the section is misleading since no transport pathways are actually analysed in this section. It is not clear how the discussion of concentration cross sections helps to explain the transport of aerosols to the Arctic without information about the simulated circulation. Typically, advection leads to import of BC mass to the Arctic in some regions and export in other regions (e.g. see Iversen, 1996).
  **Response**: Thanks for this comment. We changed the title of this section to *"PM Vertical Profiles and Associated Seasonality"*.

- P. 15, Section 4: A critical evaluation of results and comparisons with results from other studies is missing. Contributions of emissions from different geographic regions and emission sectors have been studied before (see references in introduction and review comments above). It is not obvious whether the current study adds any significant new results?

- **Response**: Previous studies have quantified the contributions from different sectors and regions using various approaches; however, a quantitative inter-comparing between our results and previous results is challenging due to multiple reasons. First and foremost, the regions (Arctic and other regions) are defined very differently in each study. In this study the Arctic region is defined as latitudes above 60 °N, while other studies have different minimum latitude for Arctic region (varying from 60 °N to 75°N). For example, Stohl et al. 2013 study shows flaring emissions contributes to 66% of Arctic BC emission (latitudes above 66 °N) but only 33% of Arctic BC emissions if Arctic region is defined as latitudes above 60 °N. Uncertainties in emissions and different simulation time periods are other factors. In the manuscript, the results were qualitatively and quantitively compared to the previous studies whenever applicable.

  As far as the authors know, this is the first study identifying sources (source sectors and geographical source regions of PM) after (Winiger et al., 2017) study which reported the error and significant over-estimation of flaring emissions. Previous studies (cited above and in the manuscript) suggested that flaring emissions from Russia are the major component of Arctic BC, while this study shows that the focus should be shifted to mitigating the emissions from residential sector (from China and Europe), industry sector from China, and European transportation emissions.

- P. 29, Fig. 1: Please indicate the base year of the emissions. It seems that emissions due to oil and gas flaring are not accounted for? These emissions are an important source of BC in the Arctic (AMAP, 2015).

  **Response:** Thanks for your comments. Discussions about emissions have been addressed in above comments and several paragraphs on emissions are added to the manuscript. But to summarize, we have used the monthly-varying emissions available for 2008 from HTAP v2.2, in this paper. HTAP v2.2 uses EDGAR v4.3 emissions over Russia (Janssens-Maenhout et al., 2015). Emissions from oil and gas flaring from Russia, is included in EDGAR inventory but it is highly under-estimated. Huang et al. 2014 study shows that emissions from several emission sectors including gas flaring and mining are significantly under-estimated in EDGAR emission inventory. We agree that under-estimation of flaring emissions could lead to the under-estimations of surface and column concentrations over the Arctic. It should also be noted that HTAP v2.2 is the emission recommended by AEROCOM.

- P. 31, Figs. 3 and 4: These figures are confusing and need to be modified or replaced. First, very large regions with considerable meteorological and chemical variations are

covered by each individual flight track so that mean results for individual flights are not very meaningful and difficult to compare with each other. Second, what is the purpose of using a time axis? Can results for individual days be used to understand the temporal evolution of plumes of polluted air? Finally, the large amount of information in the figure is overwhelming. A much better way to present the data would be to produce plots similar to Fig. SM2, with meteorological variables plotted along flight tracks.

Furthermore, it is not obvious that a comparison of meteorological results is necessary for the validation of simulated concentrations? Perhaps these figures can be moved to the supplementary document?

- **Response:** Thanks for these suggestions. First, we have moved Fig. 3 to supplementary materials. Second, the x axis (flight dates) are used as a way to distinguish flights from each other. The concentrations along each individual flight is plotted (observation vs. simulation). Discussions about temporal evolution of plumes for each of the ARCTAS flights are not in the scope of this paper. Several previous papers dedicated to temporal evolution of the plumes along ARCTAS flights over both Arctic and non-Arctic domain. For your information, we can provide the time series concentrations plots along the flight paths for all 22 flights.

    Finally, evaluation of metrological values from the meteorological model (here WRF) results is important in any air quality modeling study as it is the driver of the underlying transport patterns. For example, using a more sophisticated representation of wet deposition and its impact on the Arctic is not useful as long as we first evaluate our meteorological inputs (RH, precipitation, T, ….). Comparison of meteorological results give us the information and details necessary for validating, interpreting, and understanding any simulated concentrations. The figure is moved to SM. We also added the following to the manuscript for more clarification:

    *"Since meteorology drives the underlying transport patterns in air quality simulations, WRF model performance was evaluated using observations from 2008 ARCTAS field campaign."*

- P. 42, Fig. 14: Why are results for dust shown in this figure? Dust has not been addressed before in the paper. In particular, it is not clear how well the model simulates dust. Furthermore, the labels in the figure are much too small and so it is not clear what longitude range is shown. Locations of Eurasia and Siberia are not clear. What are the units of the concentrations?

- **Response**: Thank you for this comment. We have modified this plot and improved the quality of it. The labels are enlarged for better clarity. The x axis labels have changed. The unit concentrations were already mentioned on each contour subplot on top right but for better clarity we also added the unit concentrations are both in Y axis

labels. The units of all the subplots are µg/m$^3$ as shown in the plots. The dust row has been removed from this figure and transferred to supplemental materials Here Siberia refers to the non-European Russia with longitude between 70°E to 180°E. The term Eurasian Arctic is replaced by European arctic to reduce confusions. European arctic is between 10°W to 70°E with latitudes higher than 65 °N.

**Technical corrections:**

- There are several grammar mistakes. A few examples are given in the following. It will be necessary to check the text for additional grammar mistakes.
  P. 5: L. 9: "been" missing. L. 10: remove "the" L. 11: "an" missing. L. 16: "detail" L. 15: "The model used..." (past tense) vs "The model includes..." L. 15: missing "a"
  **Response**: Thank you. All have been corrected in the manuscript.

- P. 7, L. 27: Redundant information. Tables 1 and Fig. SM2 were already mentioned in the previous section.

  **Response**: We removed this sentence.

- P. 8, L. 19: A reference is missing for the MAC value used in the current study.
  **Response:** The reference for the MAC value used in this study has already been mentioned earlier in the manuscript in the method section.

- P. 14, L. 25: "Concentration" instead of "magnitude".
  **Response:** Thank you. This has been implemented.

- P. 36, Fig. 8: Values on the map are mentioned but are not actually visible in the plots?
  **Response:** Thanks for this comment. The caption is corrected now.

- P. 38, Fig. 10: What is OPM2.5?
  **Response:** OPM2.5 refers to as other non-carbonaceous PM2.5 (i.e. aerosols such as fly ash and road dust) that were simulated as a single component. The following has been added to the text:
  *"OPM2.5 is the acronym for other PM2.5 and refers to other primary emitted non-carbonaceous particles with aerodynamic diameters less than 2.5 µm such as fly ash, road dust, and cement which were simulated as a single mass component in the model."*

- P. 40, Fig. 12: Abbreviations BBSI, BBEU etc. in the figure need to be explained in the caption. It is not clear what these represent.

**Response:** Thank you. The descriptions of these abbreviations are added to the captions text. The following has been added to the caption of this figure:

[revised manuscript text omitted]

**Figure SM 8: Time-series concentration and contribution of different economic sectors to BC column concentrations over the Arctic.**

[Figure]

**Figure SM 79: % contributions of different sectors from Europe and China to surface BC– annual average**

[Figure]

**Figure SM 810: Seasonality % contributions of different economic sectors from China to surface BC.**

[Figure]

**Figure SM 911:** Seasonality % contributions of different economic sectors from Europe to surface BC. MAM denotes the average for months of March, April, and May. JJA denotes the average for months of June, July, and August. SON (bottom right panel) denotes average for months of September, October, and November. DJF denotes the average for the months of December, January, and February.

[Figure]

**Figure SM 12: Cross-section at 65 °N for sulfate (left) and dust (right) at different seasons MAM denotes the average for months of March, April, and May. JJA denotes the average for months of June, July, and August. SON (bottom right panel) denotes average for months of September, October, and November. DJF denotes the average for the months of December, January, and February.**

---

## Author Response (AR2)

We would like to thank you and the reviewers for your time and the constructive comments. Below is the point by point response to comments of reviewers. The reviewers' comments are shown in gray, the responses are shown in black plain text. The modified or added text in the manuscript is shown in *italics* with quotation marks.

**Response:** Thank you for your comments that have greatly improved the quality of this manuscript.

Perhaps it would be useful to clarify what is shown in figures 9, 10, and 11. In particular, the authors indicate that transport of aerosol into the model domain is simulated using specified boundary conditions. How does this approach affect results shown in these figures? It does not seem obvious how contributions of different emission sectors and regions to local aerosol concentrations in the figures can be diagnosed for aerosol transported into the model domain? The contribution of aerosols from outside of the particular model domain used in this study is not negligible, at least for some global models, i.e. models used by AMAP (2015). If this contribution is omitted in the analysis then this needs to be explained and quantified in the manuscript because otherwise results in the figures are incomplete and misleading.

**Response:**

The STEM model used fixed boundary conditions for these annual simulations. These boundary conditions varied spatially and vertically based on observations from previous aircraft field experiments and discussed in detail in Tang et al., 2004. However, emission perturbation simulations were not performed on the BC coming from the boundaries. Therefore, the contribution of BC coming from outside of the modeling domain (Southern America, Australia, and Southern Africa) is not calculated, since these emissions are not expected to have a significant contribution to the Arctic region. Previous studies have shown that the emissions from the regions outside of our modeling domain (Southern America, Australia, Central and Southern Africa) have insignificant contributions to the Arctic BC burden. For example, Reddy and Boucher, 2007 shows that emission from Australia, South America, and Africa each contribute to ~1% of sum of both Arctic and Antarctic BC surface deposition (Table 1 and 2 of

Reddy and Boucher, 2007). Similarly, Zhang et al., 2015 shows that contributions of emissions from Australia and South America contribute to 0% and <0.1% of sum of Canada, former Soviet Union and Europe BC burden. In Zhang et al. 2015 study Arctic is not defined as a separate region and it is part of Soviet Union, Europe, and Canada (Figure1 Zhang et al. 2015). Wang et al., 2014 study shows the total contributions of emissions in Southern Hemisphere to the Arctic BC column burden is <1% (Figure 6 of Wang et al. 2014).

We would like to point out that our results are in agreement with AMAP, 2015 multi-model study. In AMAP 2015, emissions from Middle East, Central Asia, Africa, South Asia and all emissions from Southern Hemisphere are lumped into Rest of the World emission (ROW) (AMAP 2015, Figure 5-1). However, our modeling domain was extended over northern Africa, Middle East, central Asia and south Asia to include the emissions from densely populated regions.  Our modeling study shows that sum of contributions from Middle East, Central Asia, and South Asia (both anthropogenic and biomass burning) to the Arctic is ~9%. This is similar to the results of AMAP 2015 showing total contributions from ROW (including the regions above) is between ~7% to ~14% (Table 11-1 AMAP 2015). Similarly (Ikeda et al., 2017) study shows that anthropogenic contributions from sources other than the four major source regions defined in the study ( Europe, East Asia, North America , and Russia) contribute to ~3% of surface BC concentration and 11% of BC concentration below 5km.

For better clarifications this paragraph is added to the manuscript:
*"It should be noted that emission perturbation simulations were not performed on the BC coming from the boundaries and the contributions of BC coming from outside of the modeling domain are not calculated since these emissions are not expected to have a significant impact on the Arctic region. Previous studies have shown that the emissions from regions outside our modeling domain (i.e. Southern America, Australia, Central and Southern Africa) have insignificant contributions to the Arctic BC burden (Reddy and Boucher, 2007; Wang et al., 2014; Zhang et al., 2015). For example, Reddy and Boucher, 2007 shows that emissions from Australia, South America, and Africa each contribute to ~1% of sum of both Arctic and Antarctic BC surface deposition. Similarly, Wang et al., 2014 study shows the total contributions of emissions in the Southern Hemisphere to the*

*Arctic BC column burden is <1%. Zhang et al., 2015 also indicates that contributions of emissions from Australia and South America contribute to 0% and <0.1% of the sum of Canada, former Soviet Union and Europe BC burden."*

In addition, it would be good to clarify the meaning of the aging rate in line 26. The units are not consistent with a rate.

**Response:** Thanks for this comment. The unit for aging rate is now corrected to $s^{-1}$.

Secondly, median values seem to be missing for some of the boxes in Figure 4.

**Response:** Thanks for this comment. Figure 4 is updated and corrected now.

[revised manuscript text omitted]

---

## Author Response (AR3)

We would like to thank you again for your helpful comments. The editor's comments are shown in gray, the responses are shown in black plain text. The modified or added text in the manuscript is shown in *italics* with quotation marks.

Thank you for the revised manuscript.

However, it would be useful to include 1 sentence in the new text you included about the fact that you find similar "ROW" contributions as in the AMAP (2015) study. You mention this in your reply but not in the text you included. Your paper is now accepted for publication.

**Response:** Thanks so much for this insightful comment. We have added the following text to the manuscript (P19) to address your comment.

[revised manuscript text omitted]

**Arctic Surface Concentration(μg/ m³ )**

[Figure]

**Figure 10: Time-series concentration and contribution of different sector to BC concentration (top panel) different sectors to PM2.5 concentration (middle panel), and different PM2.5 species (bottom panel). OPM2.5 is the acronym for other PM2.5 and refers to other primary emitted non-carbonaceous particles with aerodynamic diameters less than 2.5 μm such as fly ash, road dust, and cement which were simulated as a single mass component in the model.**

[Figure]

**Figure 11: Spatial distribution of source region contributions (%) to annual BC surface concentration over the entire domain. This figure is generated using NCAR Command Language (NCL) version 6.3.0, open source software free to public, by UCAR/NCAR/CISL/TDD, http://dx.doi.org/10.5065/D6WD3XH5.**

[Figure]

**Figure 12: Top two subplots show the seasonality of BC major geographical contributors to the Arctic (latitudes > 60 °N)surface (first row) and column (second row) concentrations. The bar plots (bottom 4 subplots) indicate the seasonality and annual average of contributions of various economic sectors from**

Europe or China to the Arctic (latitudes > 60 °N ) surface (third row) or column (bottom row) BC concentration.  BBSI, BBEU, BBNA, and BBSA denote biomass burning from Russia, Europe, North America, and South Asia respectively.  EUR and CHI denote Europe and China. Industry, power, residential and transportation sectors are represented with IND, POW, RES, and TRA acronyms. MAM denotes the average for months of March, April, and May. JJA denotes the average for months of June, July, and August. SON (bottom right panel) denotes average for months of September, October, and November. DJF denotes the average for the months of December, January, and February.

[Figure]

**Figure 13: Summary of annual mean contributions to BC, sulfate (SO₄), and PM₂.₅ by source sectors (top row) , and source regions (2nd row) at Alert, Barrow, and over the Arctic regions. The bottom two rows of bar plots show the relative contributions of various economic sectors from either China (3rd row) or Europe (bottom row) to total China or Europe contributions to Arctic BC, sulfate (SO₄) , and PM₂.₅ concentration. BB denote biomass burning in this figure.**

[Figure]

Figure 14: Cross-section at 65 °N for different seasons. The top row shows the BC concentrations (in *μ*g/m3) at the 65 °N cross-section. The 2nd, 3rd and 4th rows show the contributions of China, Europe and biomass burning (BB) to BC at 65 °N. JJA denotes the average for months of June, July, and August. SON (bottom right panel) denotes average for months of September, October, and November. DJF denotes the average for the months of December, January, and February. This figure is generated using NCAR Command Language (NCL) version 6.3.0, open source software free to public, by UCAR/NCAR/CISL/TDD, http://dx.doi.org/10.5065/D6WD3XH5.